# Finding the best trade-off between performance and interpretability in predicting hospital length of stay using structured and unstructured data

Franck Jaotombo[1,2]*, Luca Adorni[3], Badih Ghattas[4], Laurent Boyer[2,5]

**1** EMLYON Business School, Ecully, France, **2** Research Centre on Health Services and Quality of Life, Aix Marseille University, Marseille, France, **3** Becker Friedman Institute, Chicago, IL, United States of America, **4** Aix Marseille University, CNRS, AMSE, Marseille, France, **5** Department of Public Health, Assistance Publique–Hopitaux de Marseille, Marseille, France

\* jaotombo@em-lyon.com

## Abstract

### Objective

This study aims to develop high-performing Machine Learning and Deep Learning models in predicting hospital length of stay (LOS) while enhancing interpretability. We compare performance and interpretability of models trained only on structured tabular data with models trained only on unstructured clinical text data, and on mixed data.

### Methods

The structured data was used to train fourteen classical Machine Learning models including advanced ensemble trees, neural networks and *k*-nearest neighbors. The unstructured data was used to fine-tune a pre-trained Bio Clinical BERT Transformer Deep Learning model. The structured and unstructured data were then merged into a tabular dataset after vectorization of the clinical text and a dimensional reduction through Latent Dirichlet Allocation. The study used the free and publicly available Medical Information Mart for Intensive Care (MIMIC) III database, on the open AutoML Library AutoGluon. Performance is evaluated with respect to two types of random classifiers, used as baselines.

### Results

The best model from structured data demonstrates high performance (ROC AUC = 0.944, PRC AUC = 0.655) with limited interpretability, where the most important predictors of prolonged LOS are the level of blood urea nitrogen and of platelets. The Transformer model displays a good but lower performance (ROC AUC = 0.842, PRC AUC = 0.375) with a richer array of interpretability by providing more specific in-hospital factors including procedures, conditions, and medical history. The best model trained on mixed data satisfies both a high level of performance (ROC AUC = 0.963, PRC AUC = 0.746) and a much larger scope in interpretability including pathologies of the intestine, the colon, and the blood; infectious

**Data Availability Statement:** The MIMIC III database is freely available but requires special registration, as stated here: https://mimic.mit.edu/docs/gettingstarted/. The authors confirm they had

no special access privileges and that others may access the data in the same manner they did.

**Funding:** The author(s) received no specific funding for this work.

**Competing interests:** The authors have declared that no competing interests exist.

diseases, respiratory problems, procedures involving sedation and intubation, and vascular surgery.

## Conclusions

Our results outperform most of the state-of-the-art models in LOS prediction both in terms of performance and of interpretability. Data fusion between structured and unstructured text data may significantly improve performance and interpretability.

## Introduction

Hospital length of stay (LOS) is defined as the time interval between hospital admission and discharge during a given admission event [1]. As LOS enables a monitoring of the patients' flows within the hospital's care units and environment, it is considered as an indicator of resource consumption, cost and illness severity [1, 2]. Average length of stay (ALOS) is a macro indicator representing the average number of days patients spent in hospitals. It is the ratio between the sum of LOS for all inpatients in a year and the number of hospital stays, excluding day cases [3].

The ALOS in hospitals is also an indicator of efficiency in healthcare. Controlling for other factors, a shorter stay is likely to reduce the cost per stay and paves the way towards less expensive care settings. Longer stays suggest poor care coordination and may induce unnecessary in-hospital delays prior to rehabilitation or long-term care. Yet, some patients may be discharged too early when a longer hospital stay might have improved their conditions or reduced the likelihood of readmission. In 2019, the ALOS across the OECD countries was equal to 7.6 days [4].

One way to manage LOS is discharge planning. It is a customized individual plan designed for a patient, preparing the whole process leading to his leave after discharge, including the ongoing support in the community, and preventing readmission. Not only is discharge planning likely to reduce risks of readmission and improve patient satisfaction, it is especially instrumental in reducing LOS, thus significantly improving quality of care [5]. Indeed, whereas discharge planning may include several aspects such as inputs from allied health staff, and discussions with community healthcare providers, some of its critical contributions rely on estimating Discharge Date and Destination (DDD). Accurate prediction of DDD is directly based on the reliability of LOS prediction. Furthermore, not only do incorrect predictions jeopardize medical services and cause the dissatisfaction of patients and healthcare professionals, but they may also block and waste inpatient bed days. Conversely, accurate LOS prediction allows better resource allocation and care organization from patient admission to discharge preparation [6]. Reliably predicting LOS could be an effective way to reduce costs and prevent unnecessary extended stays conducive to acquired infections, falls, overcrowding, or medical errors [7].

A recent systematic review proposed to categorize the approaches to predict LOS into three main groups. The first included methods based on statistical modeling such as the generalized linear models (linear and logistic regression); the second covered methods based on operational research such as compartmental modeling, simulations, Markov models and phase-type distributions; the third were data mining and machine learning based methods [1]. With the advent of the "big data" era and the rising interest on electronic health records (EHR), the machine learning approach is gaining more momentum. Bacchi and colleagues [8] argue that

the assumption-free data-driven nature of machine learning would make it an optimal choice for reaching accurate prediction of LOS.

Lequertier et al. [6] offer another extensive review on the methods used to predict LOS. While they highlight that LOS is still relevant in planning bed capacity, and discharge planning is still a current matter of concern in healthcare delivery, they also stress the difficulty in identifying an optimal method due to the diversity of data sources, input variables and metrics. These shortcomings of the current LOS research are, furthermore, highlighted by Stone et al. [1]: "(. . .) the performance of a given approach will vary depending on a large number of competing factors such as the number of patients a hospital admits, a patient's diagnosis, the hospital's urban/rural location, particular procedures or processes in place and care units, etc." (p. 27), thus they suggest to work on models trained only on data systematically collected in the majority of hospitals. The authors equally stress the need to study the contribution of nursing admission data, given that the nurses spend much more time with the patients than the doctors, and are able to collect more information on the patients' social background, home situation, lifestyle habits and overall livelihood constraints. Lequertier et al. [6] further recommend 1) a transparent restitution of population selection, data sources and input variables, handling of missing data, LOS transformations, and performance metrics; 2) avoiding arbitrarily excluding outliers which impairs validity; 3) using different datasets for training the model and testing the performance, and even avoiding the pitfall of splitting the data into overly optimistic or pessimistic datasets by using $k$-cross-validation; 4) selecting metrics that account for the outcome distributions–especially in case of imbalanced datasets; 5) reporting the training time of the models; 6) using open and freely available datasets.

In clinical research, improving predictive performance is good but not nearly enough to encourage a wide adoption of ML models. Admittedly, the more sophisticated ML models such as Deep Learning (DL) may seem like black boxes [9, 10], which clinicians and practitioners may find disconcerting as they expect more interpretability. Clinicians will most likely be reluctant to welcome the achievements of these models despite the benefits their predictive abilities might bring, as the derivation leading to their results comes with a poor explicit explanation, if any. Consequently, developing systems that support explainable and transparent decisions have become prevalent [11] as eXplainable Artificial Intelligence–XAI [12]. Performance concerns the ability of a model to make correct predictions, while interpretability concerns to what degree the model allows for human understanding [13]. Models exhibiting high performance are often more complex and less transparent, while interpretable models may be more limited in performance. Exploring the trade-off between performance and interpretability is one of the main goals of XAI [14, 15].

As LOS is a quantitative variable, several studies attempt to predict its value with Machine Learning (ML) regression models. Yet from the perspective of identifying patients at risk, predicting prolonged LOS (PLOS) may be the main concern as opposed to regular LOS (RLOS) [16]. In such a case, the outcome to be predicted is categorical (binary) and the ML models to be used are classification models. This binarization process requires the choice of a cutoff point. However, there does not seem to be any consensus on the choice of the threshold [17]: some select ad hoc cutoffs such as 7 days to obtain more balance datasets [18], others use statistical criteria such as the 75th, the 90th or 95th percentiles [16, 19, 20]. It is therefore difficult to make a rigorous benchmark between the different studies predicting LOS [8].

One way of improving the performance of LOS prediction is to resort to other data types such as medical imaging or free texts (clinical notes) [8]. Free text may be collected from doctors' and nurses' clinical notes available in electronic health records (EHR), and leveraged to improve interpretability [1]. Not only can clinical notes predict different types of outputs [21–23] but they may also increase the performance of the typical structured datasets in predicting

LOS [18, 24]. Overall, their use may be a means of enhancing the trade-off between performance and interpretability [25].

In this article, we are exploring different ways of finding the best trade-off between performance and interpretability in LOS prediction by comparing results from models trained only on structured tabular data, with models trained only on unstructured clinical text data, and with models trained on mixed tabular structured and unstructured data—through data fusion.

## Methods

### Dataset

MIMIC-III (Medical Information Mart for Intensive Care III) is a large, freely and publicly available database comprising deidentified health-related data associated with over 40k patients who stayed in critical care units of the Beth Israel Deaconess Medical Center (BIDMC) between 2001 and 2012. It is maintained by the Massachusetts Institute of Technology (MIT)'s Laboratory for Computational Physiology and includes information such as demographics, vital sign measurements made at the bedside (~1 *data point per hour*), laboratory test results, procedures, medications, caregiver notes, imaging reports, and mortality.

### Inclusion criteria

The original admission table contains 58976 hospital admissions and 46520 patients. Only adult in-hospital stays were selected and included in the analyses. Hospital mortality, patients less than 18 years old, and hospital LOS less than 24 hours were also excluded, and duplicate admissions ID removed. The final dataset contains 30764 stays.

### Missing data

Amongst all the variables selected, only the quantitative variables had missing values. Most of these had less than 2.5% missing data. The two variables (*patient's weight*, *albumin min*) containing more than 20% missing values were dropped. Linear interpolation, a classic but dependable method, is used to impute the missing values [26].

### Study outcome

For each admission, the LOS is computed as the difference between the time of discharge and the time of admission. Prolonged LOS (PLOS) is statistically defined as any LOS greater than Tukey's regular boxplot upper fence [27] given by the following simple formula:

$$UF = Q_3 + 1.5 \times (Q_3 - Q_1)$$

Where *UF* represents the upper fence and $Q_1$, $Q_3$ are respectively the first and third quartiles

This cutoff was first chosen for a statistical reason. The distribution of the LOS is made of one narrow peak followed by a flat line of outliers, suggesting a binary distribution (Appendix 4 in S1 File). It is also justified for a historical reason: most of the studies on LOS use either regression or binary classification. Lastly, it is founded on public health reasoning. We assume that in OECD countries, PLOS are rare, certainly much less than 50% of the stays. In statistical terms, rare may translate as outliers, and the simplest way of computing outliers without making any distribution assumption is the Tukey's fences formula used here. Our study amounts therefore to a binary classification problem where the positive class represents the prolonged stays (PLOS = 7.28%) vs. the regular stays (RLOS = 92.72%).

## Study features

There were three types of features selected as predictors in the dataset.

**Structured static data.** They include (1) sociodemographic characteristics: ethnicity, insurance, religion, marital status, sex, age category; (2) hospitalization characteristics: admission type, admission location, previous admission within the 6 previous months, hospitalization via emergency departments, origin of patient, destination of patient. There were also (3) some clinical characteristics such as the simplified acute physiology score (SAPS II) [28], the sepsis-related organ failure assessment (SOFA) [29] and the international classification of disease (ICD9) main chapters [30]. Detailed descriptions of these variables are provided in Table 1.

**Structured dynamic data.** Some variables are time dependent and would usually be modeled as time series. However, building on previous works [31–33], for each of these variables we considered only one to three data points: the minimum, the maximum and the mean. These include (1) lab results: the rate of urea nitrogen, platelets, magnesium, albumin, and calcium as well as (2) charts events: respiratory rate, glucose level, diastolic and systolic blood pressure, body temperature and urine output (see Table 1).

**Unstructured data.** The MIMIC III original *notevents* table contains more than 2 million different clinical texts, grouped under 15 categories. In our study, we considered only the clinical discharge notes, as suggested by previous works [22, 34]. In case of multiple discharge notes per stay, they were merged into a single document.

## Finding the best trade-off between performance and interpretability

**Regarding the performance of imbalanced datasets.** Given a very imbalanced dataset, the predictions are biased in favor of the majority class (RLOS) [35]. Thus, the metrics based on the confusion matrix are not reliable as they assume a balanced dataset per default.

**Table 1. Descriptive statistics.**

| Categorical variables (Counts, Percentage) | | | |
|---|---|---|---|
| | | **PLOS** | **RLOS** |
| LOS (Binary) | - | 2239 (7.28%) | 28525 (92.72%) |
| ethnicity | White | 1649 (7.29%) | 20972 (92.71%) |
| | Black | 201 (7.31%) | 2548 (92.69%) |
| | Hispanic | 83 (7.36%) | 1045 (92.64%) |
| | Unknown | 193 (6.95%) | 2582 (93.05%) |
| | Other | 71 (9.06%) | 713 (90.94%) |
| | Asian | 42 (5.94%) | 665 (94.06%) |
| admission_type | Emergency | 1942 (7.65%) | 23435 (92.35%) |
| | Elective | 229 (4.69%) | 4654 (95.31%) |
| | Urgent | 68 (13.49%) | 436 (86.51%) |
| admission_location | Home | 1343 (7.22%) | 17250 (92.78%) |
| | Other | 896 (7.36%) | 11275 (92.64%) |
| insurance | Private | 801 (8.45%) | 8682 (91.55%) |
| | Medicaid | 281 (10.25%) | 2461 (89.75%) |
| | Medicare | 1063 (6.11%) | 16344 (93.89%) |
| | Government | 79 (9.27%) | 773 (90.73%) |
| | Self Pay | 15 (5.36%) | 265 (94.64%) |

(*Continued*)

**Table 1.** (Continued)

| | | | |
|---|---|---|---|
| religion | Undefined | 667 (6.96%) | 8912 (93.04%) |
| | Jewish | 153 (5.25%) | 2759 (94.75%) |
| | Catholic | 873 (7.58%) | 10637 (92.42%) |
| | Other | 233 (8.35%) | 2556 (91.65%) |
| | Protestant Quaker | 313 (7.88%) | 3661 (92.12%) |
| marital_status | Single | 664 (8.47%) | 7172 (91.53%) |
| | Couple | 1035 (7.03%) | 13686 (92.97%) |
| | Widowed | 229 (5.04%) | 4314 (94.96%) |
| | Separated | 191 (8.2%) | 2138 (91.8%) |
| | Unknown | 120 (8.99%) | 1215 (91.01%) |
| gender | 1-Male | 1306 (7.73%) | 15582 (92.27%) |
| | 2-Female | 933 (6.72%) | 12943 (93.28%) |
| age_cat | 45–64 Years | 942 (8.96%) | 9573 (91.04%) |
| | 65–84 Years | 814 (6.26%) | 12186 (93.74%) |
| | 18–44 Years | 378 (9.43%) | 3629 (90.57%) |
| | 85+ Years | 105 (3.24%) | 3137 (96.76%) |
| type_stay | 1-Medical | 1052 (5.85%) | 16936 (94.15%) |
| | 3-Surgical | 1183 (9.29%) | 11549 (90.71%) |
| | 2-Obstetrics | 4 (9.09%) | 40 (90.91%) |
| prev_adm | 1-No Hospitalization | 1852 (7.32%) | 23441 (92.68%) |
| | 3-At Least One With Emergency | 350 (7.05%) | 4612 (92.95%) |
| | 2-At Least One Non Emergency | 37 (7.27%) | 472 (92.73%) |
| dest_discharge | Home | 513 (3.16%) | 15745 (96.84%) |
| | Other | 1726 (11.9%) | 12780 (88.1%) |
| emergency_dpt | Yes | 2010 (7.77%) | 23871 (92.23%) |
| | No | 229 (4.69%) | 4654 (95.31%) |
| icd_chapter | Digestive System | 331 (10.53%) | 2811 (89.47%) |
| | Respiratory System | 192 (6.63%) | 2703 (93.37%) |
| | Circulatory System | 532 (4.73%) | 10707 (95.27%) |
| | Neoplasms | 203 (8.94%) | 2067 (91.06%) |
| | Injury Poisoning | 469 (9.42%) | 4509 (90.58%) |
| | Genitourinary System | 42 (5.9%) | 670 (94.1%) |
| | Symptoms Signs Ill-Defined Conditions | 12 (3.75%) | 308 (96.25%) |
| | Musculoskeletal System Connective Tissue | 55 (10.91%) | 449 (89.09%) |
| | Endocrine Nutritional Metabolic Immunity Disorders | 42 (5.89%) | 671 (94.11%) |
| | Mental Disorders | 8 (2.99%) | 260 (97.01%) |
| | Nervous System & Sense Organs | 42 (7.37%) | 528 (92.63%) |
| | Infectious Parasitic | 262 (10.16%) | 2318 (89.84%) |
| | Complications Pregnancy Childbirth Puerperium | 11 (10.58%) | 93 (89.42%) |
| | Skin Subcutaneous Tissue | 3 (2.86%) | 102 (97.14%) |
| | Congenital Anomalies | 8 (4.68%) | 163 (95.32%) |
| | Blood & Blood-Forming Organs | 15 (13.39%) | 97 (86.61%) |
| | Supp Factors Health Status | 12 (14.81%) | 69 (85.19%) |
| origin_patient | 2-Other | 1877 (7.57%) | 22920 (92.43%) |
| | 1-Home | 362 (6.07%) | 5605 (93.93%) |
| **Quantitative variables (Mean, Standard Deviation)** | | | |
| | | **PLOS** | **RLOS** |

*(Continued)*

**Table 1.** (Continued)

| Quantitative variables | | | |
|---|---|---|---|
| | age | 63.44 (33.01) | 76.54 (56.16) |
| | urea_n_min | 13.33 (9.72) | 15.66 (12.06) |
| | urea_n_max | 58.21 (36.3) | 33.79 (24.48) |
| | urea_n_mean | 31.43 (19.09) | 23.76 (16.72) |
| | platelets_min | 130.18 (89.77) | 174.74 (90.29) |
| | platelets_max | 497.18 (245.21) | 328.98 (166.98) |
| | platelets_mean | 281.76 (142.54) | 238.2 (108.32) |
| | magnesium_max | 2.73 (0.9) | 2.41 (0.88) |
| | calcium_min | 7.22 (0.81) | 7.87 (0.77) |
| | resprate_min | 7.8 (3.63) | 10.64 (3.43) |
| | resprate_max | 39.2 (10.33) | 30.44 (7.73) |
| | resprate_mean | 20.31 (3.5) | 18.94 (3.37) |
| | glucose_min | 72.79 (25.37) | 92.21 (27.3) |
| | glucose_max | 275.88 (174.07) | 283.18 (8472.03) |
| | glucose_mean | 136.07 (24.36) | 136.23 (118.92) |
| | hr_min | 60.99 (14.26) | 65.7 (12.76) |
| | hr_max | 132.43 (24.17) | 110.6 (22.15) |
| | hr_mean | 89.06 (12.39) | 84.26 (12.96) |
| | sysbp_min | 73.07 (20.37) | 86.81 (17.16) |
| | sysbp_max | 183.28 (31.39) | 160.58 (26.01) |
| | sysbp_mean | 123.69 (14.97) | 121.35 (15.21) |
| | diasbp_min | 31.66 (11.79) | 39.44 (11.48) |
| | diasbp_max | 116.18 (33.86) | 94.98 (24.0) |
| | diasbp_mean | 61.7 (9.68) | 61.23 (9.9) |
| | temp_min | 35.37 (1.03) | 35.84 (0.74) |
| | temp_max | 38.64 (0.9) | 37.78 (0.77) |
| | temp_mean | 37.06 (0.52) | 36.85 (0.48) |
| | sapsii | 37.14 (13.58) | 34.09 (12.49) |
| | sofa | 4.97 (3.41) | 3.92 (2.66) |
| | urine_min | 14.63 (36.99) | 33.69 (74.0) |
| | urine_mean | 119.59 (66.42) | 138.53 (760.16) |
| | urine_max | 711.37 (1101.96) | 689.14 (27302.45) |
| | los | 40.49 (18.94) | 8.95 (5.41) |

Adjusting the confusion matrix by selecting the best classification threshold may be a better solution but the metrics become then too specific and not easily generalizable [36]. One way of addressing this issue is by relying on metrics that are not threshold-dependent such as the Area Under the Receiver Operating Characteristics Curve (ROC AUC) or the Area Under the Precision Recall Curve (PRC AUC). This, however, presents some other downsides as these metrics are based on all thresholds, including the non-realistic ones [36]. Consequently, in this study we have used metrics that are both threshold specific (Accuracy and F1 score) and threshold all-inclusive (ROC and PRC AUC's).

**Using baseline comparisons.** Comparing performance between studies remains a challenge since the interpretation of the metrics depends on the dataset's outcome distribution. This problem may be overcome by providing the baseline associated to the datasets used in each of the relevant studies. The idea is to use the metrics' values of a random classifier as baselines. This can be accomplished under two different hypotheses: (1) we assume that a random

classifier predicts the prior distribution of the outcome (i.e., the proportion of each category—in our case 0.927 vs. 0.073), or (2) we assume a random uniform distribution wherein the distributions of each class are equal (for a binary classification = 0.500 each). Then, for each of these alternatives we posit that the predicted values are entirely unrelated to the actual values. The resulting confusion matrix is then given by the contingency table of expected values under the hypothesis of independence [37], and the corresponding Accuracy and F1 score may be used as baselines (see Appendix 1 in S1 File).

For the ROC AUC, the baseline is fixed at 0.500 [38] and simple rules of thumb may be used to decide how well-performing a model is (0.5 is bad, 0.7 is acceptable, 0.8 is good, 0.9 is excellent, 1.0 is perfect). The baseline for the PRC AUC amounts to the ratio of positive observations [39] which in our case is equal to 0.073. In sum, the performance of a ML classifier should be assessed by examining how far the value of a metric is from the baseline of the corresponding random model, but also how these values compare to previous relevant studies.

**On the role of interpretability.** In clinical research, possibly more than in other disciplines, the interpretability of the results is paramount. While it is certainly essential to be able to predict which patient or stay is most likely to lead to a prolonged LOS, it is even more important to determine which factors must be attended to as a way to prevent these risks. Thanks to the current development of the field of XAI, it is increasingly easier not only to explain the global relationship between a predictor and the outcome, but also to have a finer understanding of the behavior of each instance in the prediction process [14]. There are many resources available on XAI [40], including methods to estimate variable importance such as the Leave One Covariate Out [41]. In this study we have focused mostly on the overall importance of the 20 most relevant features, either through permutation importance [42] or through Local Interpretable Model-agnostic Explanations (LIME) [43]. In the latter case, for each feature, we computed the value of its local contribution on predicting each instance, then averaged their absolute values over the whole dataset [44].

## Comparing structured, unstructured and mixed datasets

Previous studies have highlighted how the inclusion of unstructured clinical text data may improve clinical outcome predictions in quality as well as in quantity [23] especially for prolonged ICU stays, as in our case [45]. Furthermore, it is hypothesized that unstructured text data would provide richer insights into the patients since they describe symptoms, diagnosis, history, and other relevant clinical information. However, to the best of our knowledge, few studies have convincingly demonstrated so. Additionally, it remains unclear whether the inclusion of clinical data improves only interpretability, only performance or both [25].

**Structured data.** Both static and dynamic structured data were merged as one structured tabular data and used to compare 14 ML models using AutoGluon TabularPredictor [46]. The selected ML models cover a wide range of the most current, the most relevant, and best performing pre-tuned models available per default in AutoGluon:

- Five versions of the best performing boosted trees: Catboost [47, 48], LightGBM with regular trees, extra trees or large trees [49], XGBoost [50];

- Two versions of the Random Forest using respectively the Gini or Entropy loss functions [51];

- Two versions of Extra Trees using respectively Gini or Entropy loss functions [52]

- Two versions of respectively Torch and Fastai Pretuned Feed Forward Neural Networks [53];

- Two versions of the $k$-nearest neighbors, using respectively Distance and Uniform weights [54];

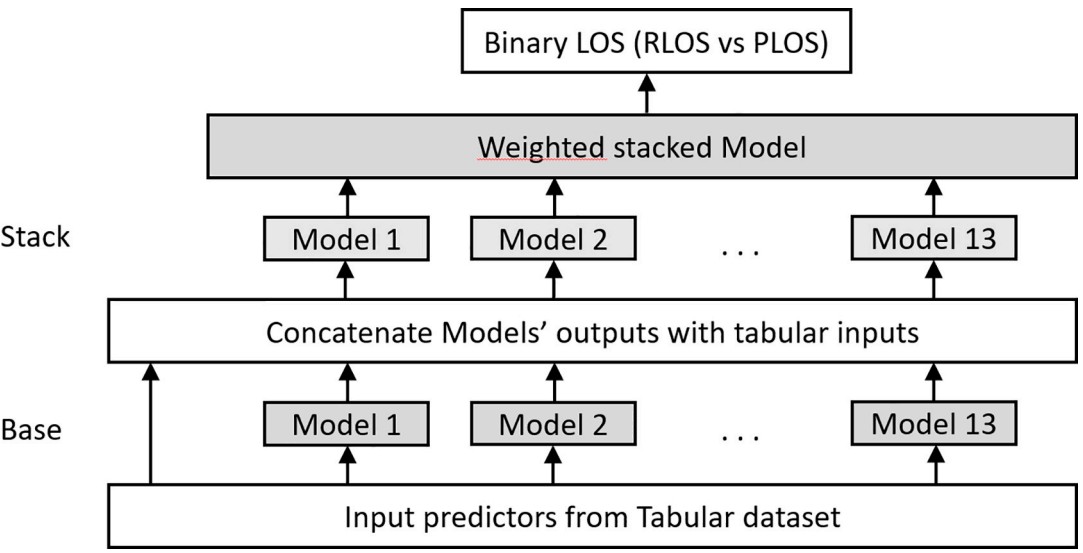

**Fig 1. Multilayer stacked ensemble (the shaded boxes are learned).** Adapted from [46].

- One ensemble learning model using a weighted stacked model of the 13 previous ML models.

Considering that the first 13 ML models make the first layer of the architecture, the stacker model takes as input not only the predictions of the models at that layer, but also the original data features themselves (input vectors are data features concatenated with lower layer model predictions). Not unlike skip connections in deep learning, this enables the higher-layer stacker to revisit the original data values during training [46]. Fig 1 summarizes the architecture of the multilayer stacked ensemble.

*Hyperparameter tuning.* The AutoGluon platform provides sophisticated means of tuning the hyperparameters. However, given the large number of models to be trained, the already satisfactory level of performance with the default parameter values, and the goal of our study, we have reduced this part to the bare minimum i.e., the choice of the evaluation metrics in tuning: the ROC AUC. We used version 0.4.1 of AutoGluon. Both parameters and models may have changed and improved over time due to the high frequency of new releases from the AutoGluon team. To properly replicate our results or check the hyperparameters in detail, the correct older version of the package must be downloaded from PyPi (https://pypi.org/project/autogluon/#history). In principle, the latest version of AutoGluon should nonetheless lead to very close if not identical results.

**Unstructured data.** As Transformers have proven to be amongst the very best models in text classification through its encoder structure [55], and since AutoGluon is Transformers' friendly, we have used its TextPredictor module to predict LOS using only unstructured text data, more precisely, the clinical discharge notes from the MIMIC III database. TextPredictor fits individual Transformer neural network models directly to the raw text.

*Hyperparameter tuning and transfer learning.* TextPredictor is capable of using pretrained models as those used by Hugging Face [56] which need only to be fine-tuned through transfer learning [57]. Since a Transformer of the BERT family [21] pre-trained on our topic is already available [58], we selected this in setting our TextPredictor hyperparameters.

**Mixed data.** There are many different ways of merging structured tabular data with unstructured text data. All of these ways, however, will require one way or another of transforming text into numbers through vectorization or embedding [59, 60]. In our study we have vectorized the clinical text data through Bag of Words (BOW) [61] followed by Latent Dirichlet Allocation (LDA) dimension reduction through topics modeling [62]. The different topics yielded by LDA

were then merged with the tabular dataset, giving rise to a new tabular dataset with its number of columns extended by the dimensions of the topic modeling vectors ($d = 300$). The AutoGluon TabularPredictor may then be leveraged as previously for the structured data. In the BOW vectorization, the document (rows) to terms (columns) or DTM matrix may use different types of occurrences' weighing. We have explored here the 3 most common ones [63]:

- **Terms Frequency** (*TF*: $weight_{i,j} = frequency_{i,j}$ *i.e frequency of term i in document j*)

- **Terms Frequency Inverse Document Frequency**
  (*TFIDF* :  $weight_{i,j} = frequency_{i,j} \times log_2 \frac{Document\_size}{frequency_i}$)

- **Binary Frequency** (*BIN*: $weight_{i,j} = 1$ *if term i is in document j*, 0 *otherwise*)

## Model training and performance estimation

To avoid overfitting, AutoGluon uses repeated $k$–*fold* bagging. It consists in randomly partitioning the data into $k$ disjoint chunks, stratified on the labels, then training $k$ copies of a model with a different data chunk held out from each copy. Applying bagging (bootstrapping then averaging over all the independent predictions from bootstrapped samples), each model is asked to produce out-of-fold (OOF) predictions on the chunk it did not see during training. This $k$–*fold* bagging process may then be repeated on $n$ different random partitions of the training data, averaging all OOF predictions over the repeated bags. The best model is obtained based on the best average validation score and a test score is computed from a test sample that was held out before model training.

## Machine learning and deep learning models

All models have been trained using Google Colab Pro+ with GPU enabled machines. Google Colab assigns a type of machine every time a new notebook is initialized, but may switch to other types. Examples of machine used are: V100 (GPU RAM: 16GB; CPUs: 2 vCPU, up to 52 GB of RAM); P100 (GPU RAM: 16 GB; CPUs: 2 vCPU, up to 25 GB of RAM); T4 (GPU RAM: 16 GB; CPUs: 2 vCPU, up to 25 GB of RAM).

## Results

### Structured data

**Performance.**   Table 2 displays the performance of the 14 models selected for the structured tabular dataset, the validation score, the holdout test score, the fit time in wall clock time, the layer where the model is located in the ensemble stacking process and their fitting order. Obviously, the ensemble learning model is at level 2 and fitted last.

Table 3 displays the overall performance of the best (weighted ensemble) model based on the 4 metrics we have selected. The AUC scores suggest a very good performance compared to the baseline values based on random models.

**Interpretability.**   Fig 2 summarizes the permutation feature importance for the best model. Results indicate that PLOS is mostly predicted by the level of blood urea nitrogen and blood platelets.

### Unstructured data

**Performance.**   Table 4 summarizes the performance of the Transformer model. The performance has dropped notably compared to the structured data, but overall these values remain well above the baseline values.

**Table 2. ROC AUC performance for the structured data.**

| model | Test score | Average validation score | Fit time (seconds) | Stack Level | For order |
|---|---|---|---|---|---|
| WeightedEnsemble_L2 | 0.944 | 0.948 | 72.914 | 2 | 14 |
| CatBoost | 0.942 | 0.941 | 2.972 | 1 | 3 |
| LightGBM | 0.942 | 0.942 | 9.102 | 1 | 7 |
| LightGBMXT | 0.940 | 0.944 | 1.252 | 1 | 4 |
| XGBoost | 0.940 | 0.940 | 2.318 | 1 | 13 |
| LightGBMLarge | 0.940 | 0.943 | 1.164 | 1 | 11 |
| RandomForestEntr | 0.938 | 0.941 | 27.369 | 1 | 12 |
| ExtraTreesEntr | 0.935 | 0.930 | 3.548 | 1 | 6 |
| ExtraTreesGini | 0.933 | 0.939 | 27.798 | 1 | 10 |
| NeuralNetTorch | 0.927 | 0.927 | 3.961 | 1 | 5 |
| RandomForestGini | 0.927 | 0.920 | 1.349 | 1 | 8 |
| NeuralNetFastAI | 0.926 | 0.927 | 1.556 | 1 | 9 |
| KNeighborsDist | 0.811 | 0.775 | 0.025 | 1 | 2 |
| KNeighborsUnif | 0.808 | 0.776 | 0.026 | 1 | 1 |

**Interpretability.** Fig 3 display the globally averaged local feature importance of each processed token from the unstructured discharge notes based on absolute weights. Information from stemming and lemmatization may be considered complementary, and the most important features in predicting PLOS are now more interpretable in terms of patient's conditions and care delivery. It appears for instance that procedures such as tracheostomy or biopsy, and conditions such as aneurysm were associated with prolonged LOS. The term "ed" refers to "Emergency Department" and, when looking at keywords-in-context for such abbreviation, it can be seen how it is frequently used when reporting vital measures taken during stay in such departments. Interestingly, the model seems to find words such as "present", "past" and "history" as highly predictive–potentially implying that Bio Clinical BERT picks up evidence of medical history and recognize it as important for evaluating the health conditions of the patients. A reliable interpretation, however, should include examination of keywords in context. For instance, the token "1" may appear as noise. Its presence may be explained by our choice of a light preprocessing, where we avoided removing numbers to preserve potentially important information (*e.g. medication quantities*). When looking at the most common keywords in the context of such token, we do in fact find a variety of medication-related words, such as "sig", "mg", "tablet", "capsul", "daili", "po", implying that Bio Clinical BERT utilizes it to spot medication frequency or dosage. It is important to consider that the LIME representations used here are based on linear (Lasso) approximations from two different models, each using a different type of preprocessing (respectively lemmatization and stemming). Each token should therefore be interpreted in light of their covariate tokens. As shown in the Fig 3, the

**Table 3. Performance metrics for the weighted ensemble model.**

| Metrics | performance | baseline |
|---|---|---|
| PRC AUC | 0.655 | 0.073 |
| ROC AUC | 0.944 | 0.500 |
| Accuracy | 0.947 | 0.865 [0.500] |
| F1 score | 0.538 | 0.073 [0.127] |

Accuracy and F1 score values in square brackets are based on uniform distribution of outcome.

The other Accuracy and F1 score baseline values are based on outcome prior distribution.

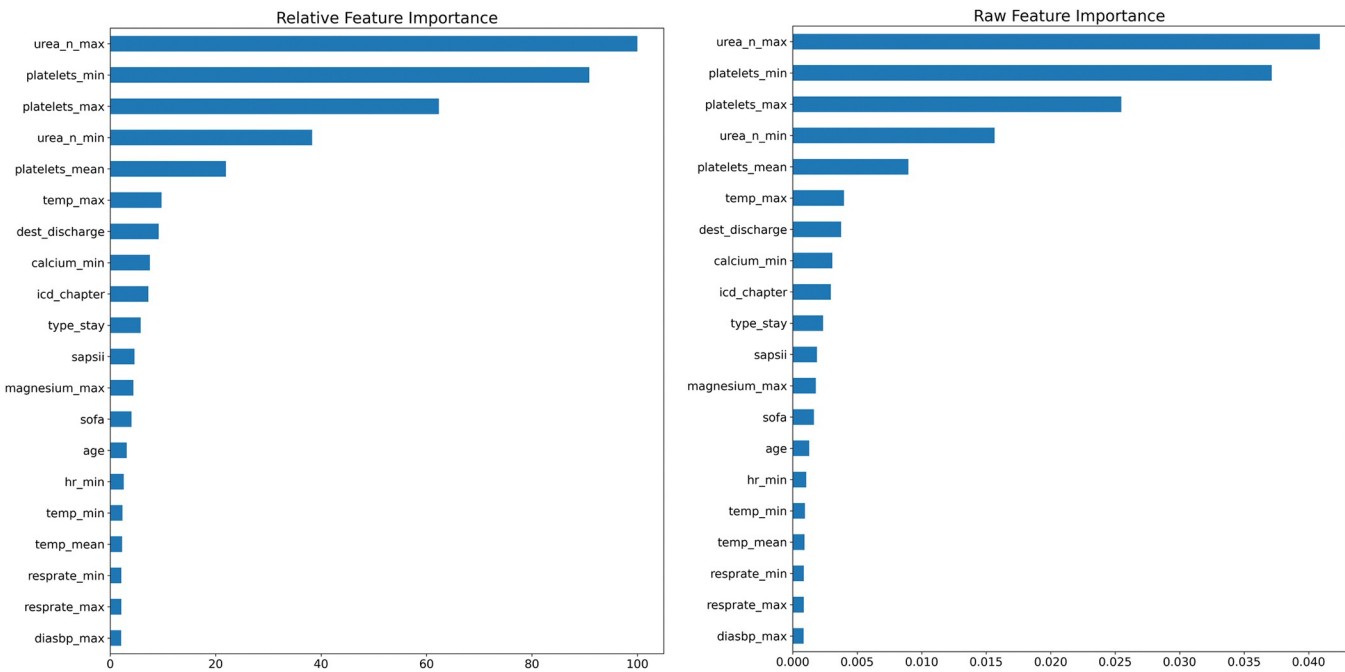

**Fig 2. Permutation feature importance for the weighted ensemble model.**

most important tokens in each model are not the same. This is a compelling evidence that pre-processing matters and that each token should be interpreted within its context.

## Mixed data

**Performance.** Table 5 summarizes the performance of the best model in each type of data preprocessing (stemming vs. lemmatization) and in each type of occurrence's weighing in the DTM table.

The performances on the mixed data are comparable to that of the structured data, i.e., very good compared to the baseline, with a notable increase in PRC AUC performance. The Term Frequency weighing with lemmatization preprocessing stands out above all in terms of AUC scores.

**Interpretability.** As we can see in Table 6, for each type or occurrence's weighing, the 20 most important features include one or several emerging topics from LDA, each identified by F followed by a number. Each topic may have been derived subsequently to a stemming or a lemmatization.

**Table 4. Performance of bio clinical BERT on the unstructured data.**

| performances | stemming | lemmatization | baseline |
|---|---|---|---|
| PRC AUC | 0.364 | 0.375 | 0.073 |
| ROC AUC | 0.839 | 0.842 | 0.500 |
| Accuracy | 0.924 | 0.921 | 0.865 [0.500] |
| F1 score | 0.337 | 0.386 | 0.073 [0.127] |

Accuracy and F1 score values in square brackets are based on uniform distribution of outcome.

The other Accuracy and F1 score baseline values are based on outcome prior distribution.

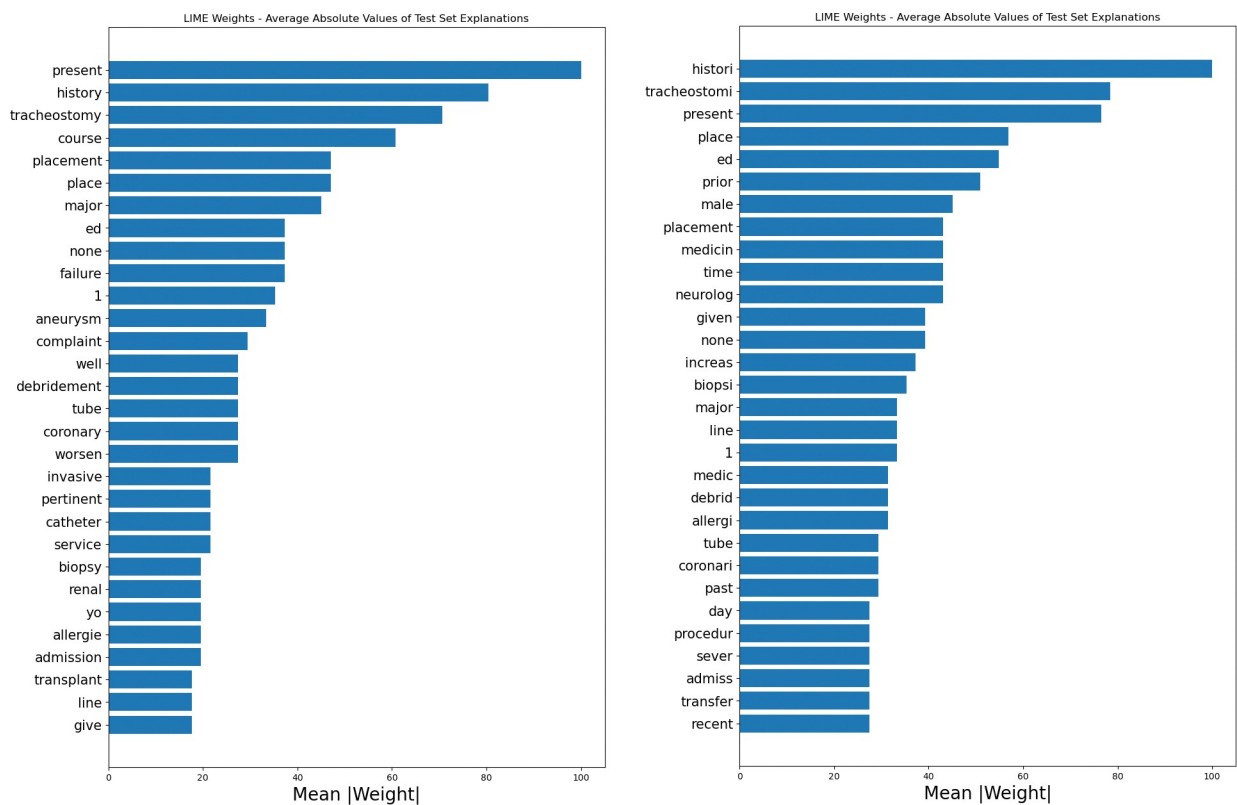

**Fig 3.** Averaged local (LIME) absolute values feature importance for the BERT Transformer (left: lemmatization, right: steming).

Tables A2.i and A2.ii in the Appendix (S1 File) provide the first 20 tokens belonging to each LDA topic, lending more contents, contexts, and descriptions of the stays at risk of PLOS.

1. **TF vectorization topics**

- F240: is referring mostly to **respiratory problems, intubation, and sedation** (tokens = *intubate, extubate, sedate, endotracheal tube, respiratory failure*)

- F268: is related to intensive care and **peg (percutaneous endoscopic gastrostomy)** related procedures (tokens = *tube feed, nutrition, drainage, peg*)

- F274: mixes part of the tube feeding procedures from F268 with **intubation** to facilitate **respiration** (tokens = *tube, tube feed, tracheostomi, ventil, respiratori*)

**Table 5. Performance of the best models for mixed data.**

|  | TF | | BIN | | TFIDF | | |
|---|---|---|---|---|---|---|---|
|  | stemming | lemmat. | stemming | lemmat. | stemming | lemmat. | baseline |
| **PRC AUC** | 0.741 | 0.746 | 0.710 | 0.701 | 0.673 | 0.690 | 0.073 |
| **ROC AUC** | 0.961 | 0.963 | 0.957 | 0.957 | 0.951 | 0.954 | 0.500 |
| **Accuracy** | 0.956 | 0.956 | 0.953 | 0.951 | 0.947 | 0.950 | 0.865 [0.500] |
| **F1 Score** | 0.633 | 0.633 | 0.602 | 0.598 | 0.536 | 0.561 | 0.073 [0.127] |

Accuracy and F1 score values in square brackets are based on uniform distribution of outcome.

The other Accuracy and F1 score baseline values are based on outcome prior distribution.

**Table 6. Permutation feature importance for the best models on mixed data.**

| TF | | | | | | BIN | | | | | | TFIDF | | | | | |
|---|---|---|---|---|---|---|---|---|---|---|---|---|---|---|---|---|---|
| Stemming | | | Lemmmatization | | | Stemming | | | Lemmmatization | | | Stemming | | | Lemmmatization | | |
| urea_n_max | 100 | 1 | urea_n_max | 100 | | urea_n_max | 100 | 1 | urea_n_max | 100 | | urea_n_max | 100 | 1 | urea_n_max | 100 | |
| platelets_max | 68.93 | 2 | platelets_max | 63.04 | | platelets_max | 59.03232 | 2 | F258 | 91.10 | | platelets_min | 75.17 | 2 | platelets_min | 86.60 | |
| urea_n_min | 49.20 | 3 | platelets_min | 49.98 | | platelets_min | 55.5567 | 3 | platelets_max | 69.76 | | platelets_max | 56.28 | 3 | platelets_max | 66.70 | |
| platelets_min | 48.68 | 4 | urea_n_min | 43.61 | | urea_n_min | 49.51 | 4 | platelets_min | 54.13 | | urea_n_min | 46.14 | 4 | urea_n_min | 47.10 | |
| F274 | 19.39 | 5 | F268 | 19.63 | | F99 | 31.70 | 5 | urea_n_min | 51.92 | | F1 | 13.85 | 5 | F59 | 24.25 | |
| F87 | 14.98 | 6 | F240 | 9.10 | | F43 | 27.39 | 6 | F27 | 18.68 | | type_stay | 11.12 | 6 | F205 | 17.42 | |
| F195 | 10.06 | 7 | temp_max | 8.87 | | F27 | 11.96 | 7 | type_stay | 13.85 | | platelets_mean | 10.73 | 7 | platelets_mean | 10.68 | |
| temp_max | 9.87 | 8 | F180 | 6.30 | | type_stay | 10.60 | 8 | temp_max | 11.51 | | temp_max | 9.60 | 8 | temp_max | 9.82 | |
| F32 | 9.66 | 9 | temp_min | 5.79 | | F255 | 9.13 | 9 | F184 | 9.93 | | F123 | 5.71 | 9 | type_stay | 8.75 | |
| F242 | 4.74 | 10 | F88 | 5.71 | | temp_max | 9.07 | 10 | F81 | 9.56 | | calcium_min | 5.37 | 10 | F198 | 7.11 | |
| type_stay | 4.56 | 11 | platelets_mean | 5.69 | | sofa | 7.80 | 11 | sapsii | 9.41 | | sapsii | 4.94 | 11 | dest_discharge | 6.20 | |
| platelets_mean | 4.47 | 12 | F60 | 5.58 | | temp_min | 6.60 | 12 | sofa | 7.98 | | sofa | 4.40 | 12 | F122 | 5.80 | |
| temp_min | 4.14 | 13 | F174 | 4.67 | | platelets_mean | 5.92 | 13 | F232 | 7.44 | | F162 | 4.36 | 13 | sofa | 5.18 | |
| resprate_max | 4.08 | 14 | type_stay | 4.65 | | calcium_min | 5.63 | 14 | F163 | 7.37 | | F57 | 4.30 | 14 | F123 | 4.75 | |
| sofa | 4.01 | 15 | F16 | 4.53 | | sapsii | 5.57 | 15 | F284 | 7.12 | | F60 | 3.86 | 15 | F233 | 4.32 | |
| F92 | 3.63 | 16 | F101 | 4.27 | | F286 | 5.48 | 16 | temp_min | 7.01 | | icd_chapter | 3.52 | 16 | sapsii | 4.14 | |
| F219 | 3.60 | 17 | F87 | 3.54 | | F210 | 5.39 | 17 | F80 | 6.60 | | F21 | 3.15 | 17 | temp_min | 4.08 | |
| urine_min | 3.46 | 18 | glucose_max | 3.30 | | F12 | 4.23 | 18 | platelets_mean | 5.71 | | dest_discharge | 3.04 | 18 | calcium_min | 3.52 | |
| sapsii | 2.98 | 19 | F86 | 3.24 | | F141 | 3.55 | 19 | F279 | 5.52 | | age | 3.00 | 19 | magnesium_max | 3.36 | |
| F294 | 2.95 | 20 | F251 | 3.18 | | F202 | 3.48 | 20 | F277 | 5.50 | | temp_min | 2.72 | 20 | F121 | 3.03 | |

- F87: focuses on **drainage procedures** related to the abdomen (tokens = *fluid collect, abscess, drainage, cathet*)

2. **Binary vectorization topics**

- F258: similarly to the Term Frequency topics, it evokes **intubation and assisted feeding** (tokens = *tube feed, surgery, intensive care, drain*)

- F27: is related to patients affected by **cancer**, describing both its diagnosis and subsequent therapy (tokens = *biopsy, cancer, metastatic, mri, chemotherapy, oncology, tumor, malignancy*)

- F99: is clearly related to **infectious diseases** and related medication (tokens = *infecti diseas, antibiot, vancomycin, zosyn, flagyl*)

- F43: evokes **treatment of wounds post operations** (tokens = *wound, dress chang, tissue, drainage, surgery, tissue, heal*)

3. **TFIDF vectorization topics**

- F59: **pathology of the colon** with potential surgery and complications or with external evacuation in a bag (tokens = *ostomy, ileostomy, laparotomy, abscess, drain, fluid collection, tpn, fistula, adhesion*) with also mentions of methods of artificial feeding–TPN, Total Parenteral Nutrition

- F205: is mostly **treatment** related with words evoking either medication frequency, intensive care and clinical measures (tokens = *mg po, hematocrit, care unit, blood pressure, intensive care, rate, pressure*)

- F1: similarly to F205, contains a host of **medical abbreviations for daily medications**, such as *mg po*, indicating quantity (*mg*) and assumption method (*po = per os*, *i.e.*, *by mouth*) or *po qd*, indicating daily oral consumption

- F123: **pathology of the blood** (tokens = *bone marrow, lymphocyt, leukemia, lymphoma, chemotherapi, neutropenia*) generally associated with cancer. This is not only a confirmation of the results given by the structured data but expounds on it through information on the conditions and the type of disease related to platelet counts.

To summarize, certain conditions appear to be risk factors for PLOS whereas others appear to be mitigating factors. The first category includes pathologies of the intestine and the colon, pathologies of the blood and infectious diseases, respiratory problems, and lastly treatment and diagnosis of cancer cells. It is also related to conditions requiring sedation, intubation and artificial feeding.

The second category includes continuity of healthcare delivery, positive signs in medical auscultation, and continuous treatments, such as in topics related to the cleaning of wounds (F43) and topics connected to proper and continuous medication (F1, F205).

## Discussion

Recent systematic reviews have stressed that providing accurate predictions of Hospital Length of Stay (LOS) remains a current issue as is planning bed capacity, and patient discharge remains a serious matter in healthcare delivery [6]. These authors also highlight the need to include a transparent restitution of population sample selection, data sources, and input variables, as well as data cleaning and preprocessing procedures such as imputation strategies for missing data, LOS modeling format with potential transformations, LOS prediction methods, validation study design, and performance evaluation metrics.

These issues have been addressed here in various ways. The criteria of inclusion are clearly provided, a full description of the data sample is provided in Table 1 and the missing data imputations made explicit. The rationale for binarizing the LOS output variable is explained and the code containing the whole preprocessing of the dataset along with all the code used in the study are openly available in the GitHub of the study (link: https://github.com/jaotombo/LOS_mixed_2022). The choice of the evaluation metrics is clearly outlined and justified, and the validation design made explicit and justified with the proper citations. Furthermore, a separate holdout test set was used in addition to *k–fold* cross-validated sets and multiple resamplings (repeated *k–fold* bagging) [46]. Information on the training time of the models is also provided to meet the requirement of digital resources minimization. Lastly, the use of open and freely available datasets has been adopted to facilitate benchmarking, replication, and external validity.

We agree with these authors' recommendation in adopting metrics agnostic to the outcome distribution—such as the AUC. However, to facilitate benchmarking between studies using different datasets and outcome distributions, we additionally suggest that researchers present the baselines adopted as references for the metrics selected. We recommend constructing these baselines from the performance of an appropriately defined random model. For example, such a random model may predict classes based on a uniform distribution or on prior probabilities of the outcome classes (Appendix 1 in S1 File). The performance of each model is then ascertained (as an absolute difference) from these baselines. From this perspective, threshold-based metrics remain useful and informative.

Several other studies have suggested the use of natural language data as a way to improve performance amongst which are Bacchi et al. [8] in their systematic study of LOS or Shickel

et al. [25] in their survey of deep learning techniques used in electronic health records (EHR). The latter affirms that clinical text data are perhaps "the most untapped resource for future deep clinical methods" as it "contains a wealth of information about each patient" [25]. This observation is especially born from the concern to secure more interpretability in Machine and Deep Learning models. Hence, more and more studies set out to predict health outcomes using unstructured clinical text data [21–23].

We have found few studies predicting LOS with unstructured text data. The MIMIC III dataset was used to predict a composite outcome *Hospital Death = Yes or LOS ≥ 7 days* [45] comparing only structured data and structured + unstructured text data (ROC AUC score are respectively equal to 0.83 & 0.89 for their best model—Gradient Boosting). This study not only displays a high level of performance, but it also provides some elements of interpretability, albeit somewhat limited as it used the logistic regression's odd ratios to assess variable importance. That these assessments remain applicable to their best model is not warranted. Another study [18] first processes the clinical text data through the Unified Medical Language System (UMLS) then uses the 969 concepts extracted thereof as a new set of categorical variables to be included in their model through one-hot encoding. The authors define PLOS as *LOS ≥ 7 days* and obtain a balanced dataset, justifying the use of Accuracy and F1 as metrics. Still, their best performance (F1 = 0.875) is to be assessed with a baseline of a balanced dataset (= 0.500 –see Appendix 1 in S1 File) while our best model displays a F1 = 0.633 with a baseline of 0.073 [0.127] (Table 5), so on F1 score our model is superior. On Accuracy, their best score is 0.763 (baseline = 0.500) compared to ours: 0.956 (baseline = 0.865 [0.500]) hence, their model is better on the prior distribution baseline but ours is better on a uniform distribution baseline. Interpretability is examined through relative feature importance of the Random Forest, and this study also compares models trained from structured data only with models including both structured and unstructured data. On F1 Score and Accuracy, the models trained on mixed data are performing better than the alternatives. Limitations of this study include the difficulty to compare with other studies using AUC metrics and its inability to provide a richer interpretable information from its unstructured data.

Another study comparable to ours is that of Zhang et al. [24] which explicitly compared structured, unstructured and mixed data from the MIMIC III dataset, using both classical baseline Logistic Regression and Random Forest models, on the one hand, and different ways of merging structured and unstructured data, on the other hand, with deep learning models (Convolutional Neural Networks = CNN and Long Short-Term Memory Recurrent Neural Network = LSTM RNN). Furthermore, the authors explore 3 different outcomes: in-hospital mortality, 7 days prolonged LOS, and 30 days hospital readmission. The metrics used are the F1, the ROC AUC and the PRC AUC. Given their selected cut point on LOS, their model is well balanced (49.9% PLOS vs 50.1% RLOS).

Overall, Zhang et al. [24] best performance is given by the CNN model trained on mixed data (F1 = 0.725 (baseline 0.500)–PRC AUC = 0.662 (baseline 0.500)–ROC AUC = 0.784) whereas our best performance yields F1 = 0.633 (baseline 0.073 [0.129])–PRC AUC = 0.746 (baseline 0.073)–ROC AUC = 0.963. Compared to the baselines, our mixed model with a fusion between the structured data and the LDA vectorized unstructured text data is therefore distinctly more performant.

Beyond our model's performance, its strongest contribution may be in the interpretability of the results. Some studies confirm that a high rate of urea nitrogen is associated with PLOS in intensive care units (ICU) [64] or with a higher mortality risk due to pulmonary embolus [65], or also with elder patients [66]. Conversely, a low platelets count is associated with PLOS due to higher infectious risks [67] or to post-surgery complications [68, 69]. These results have mostly been retrieved in a single stroke by our mixed data-trained best model (Table 6). Not

only are we able to determine that the rate of urea nitrogen and platelets are the strongest predictors of a prolonged hospital stay (PLOS), but we are also in a position to describe with rich details the profile of the stays or patients at risk of PLOS. Indeed, those who are more at risk have pathologies of the intestine, or of the blood. Infectious diseases and conditions requiring sedation and intubation are also risk-prone, as are cancer affected patients.

Our results also suggest ways to mitigate these risks amongst which is a well-planned continuity of care from the moment of admission, during the whole stay, to the period after discharge. Regular treatment, medication intake, and medical auscultation are also mitigating factors.

There may be several applications to models like ours. They may be utilized as tools to aid making a precise diagnosis leading to highly desirable personalization of patients' management [70]. Better adapted to big data than the conventional statistical models, they may scale to include up to billions of patients' records, and use a single, distributed patient representation–from different data sources such as EHRs, genomics, social activities and other features describing individual status. Deployed into a healthcare system, these models would be constantly updated to follow the changes in patient population and will support clinicians in their daily activities [71]. Another area where these models may have comprehensive leverage is in healthcare operations management. ML models based on weak learners such as in boosted models or in ensemble learning models have shown to be quite relevant in predicting workflow events as well as in identifying key operational features [72]. Their efficacy is substantiated by acknowledging that any outcome of a clinical workflow is influenced by a plethora of different factors, and each of them can be considered as a weak learner due to their little impact on the outcome. As an illustration, one boosted ML model, deployed on an information system and trained in real time was used to predict waiting time in a facility, and hailed by the patients [72]. A different display was also made available and customized as an administrator view for the facility manager, allowing the staff to examine gaps between the actual and the predicted values, and providing the means to investigate new features to be used for improvement. As the performance of the models reach a satisfactory level, feature selection such as retaining the most important features were applied to determine the key factors contributing to the operational outcome—e.g., time delay in the creation of radiology reports [72]. It is not too much of a stretch to envision how these different applications would be enhanced–in terms of performance and explainability–if fused structured and unstructured data were used to train the ML models. It would improve the patients' journey, support the practitioners in their monitoring and caring tasks, and facilitate the (resources) planning and management of the facilities.

This study is not without limitations. The MIMIC III database is used here in a retrospective study. In real life and in real time, many of these variables will not always be available, thereby questioning the generalizability of our results. For the sake of generalizability, one should favor those variables that are primarily, systematically or routinely collected in most hospitals [1, 72]. Our results are very specific to the Boston Beth Israel Deaconess Medical Center with a focus on intensive care units; thus it may not generalize well. Yet, keeping only those variables routinely collected in most hospitals will reduce performance and interpretability as it will not include relevant variables that are specific to each institution and conducive to greater performance and interpretability. The specificity of each hospital may be accounted for through usage of ready to use models retrained on each local site data through threshold adjustment and transfer learning [73]. One may argue that if generalizability may be a priority for research, including all pertinent data in the model as to maximize performance and interpretability may be the priority for the practitioners and the managers. Indeed some authors recommend to embrace a wider view of generalizability where the goal is to focus on

broader questions about when, how, and why ML systems have clinical utility and ensure that these systems work as intended for both clinicians and patients [74].

Furthermore, we have limited our exploration of the unstructured text data to the discharge notes only, so many more different clinical text data have not been accounted for and may provide critical information for prediction and interpretation. Additionally, AutoGluon offers the possibility to use a multimodal format as an alternative for data fusion between structured and unstructured text data. While this approach can notably improve performance, unfortunately, in the current state of affairs, it does not seem to tap into the rich interpretability of the text. Lastly, there are other means to explore interpretability beyond LIME, amongst which are SHAP and the Shapley Values. We have found these approaches to be impractical with our dataset given the excessively long computing time—a downside acknowledged by other authors [75]. Future studies will explore these issues more in depth.

## Supporting information

**S1 File. Appendices.**
(DOCX)

**S2 File. Supplementary material.**
(DOCX)

**S3 File. Supporting information.**
(DOCX)

## Author Contributions

**Conceptualization:** Franck Jaotombo.

**Data curation:** Luca Adorni.

**Formal analysis:** Franck Jaotombo, Luca Adorni, Badih Ghattas.

**Investigation:** Franck Jaotombo.

**Methodology:** Franck Jaotombo.

**Resources:** Laurent Boyer.

**Software:** Luca Adorni.

**Supervision:** Franck Jaotombo, Laurent Boyer.

**Writing – original draft:** Franck Jaotombo.

**Writing – review & editing:** Franck Jaotombo, Luca Adorni, Badih Ghattas, Laurent Boyer.

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
