## [Decision Letter · Decision Letter 0]

9 May 2023

PONE-D-23-09830Finding the best trade-off between performance and interpretability in predicting Hospital Length of Stay using Structured and Unstructured DataPLOS ONE

Dear Dr. Jaotombo,

Thank you for submitting your manuscript to PLOS ONE. After careful consideration, we feel that it has merit but does not fully meet PLOS ONE’s publication criteria as it currently stands. Therefore, we invite you to submit a revised version of the manuscript that addresses the points raised during the review process.

1. Please add to the abstract that your experiments were performed using AutoGluon if there is no novelty in the machine learning techniques (e.g., development of a new transformer) in your methods.

2. The text in figures 1-3 is blurred. Please replace them with figures of appropriate resolution.

3. For Figures 2 and 3, instead of showing the absolute values of the weights, please show the important words in the top 20 positive and negative values.

4. Please provide visual representations of attention (similar to figures 3 and 4 in "Attention Is All You Need", arXiv:1706.03762) to predict dwell time.

5. Please make the source code and relevant tutorials/examples available as part of an open source repository on GitHub or GitLab. In addition, include a link to the repository in the manuscript.

6. Clarify authorship (https://journals.plos.org/plosone/s/authorship)

Please respond to all comments and concerns raised by the reviewers before submitting a revised version of the manuscript.

We look forward to receiving your revised manuscript.

Kind regards,

Takuma Shibahara

Academic Editor

PLOS ONE

Journal Requirements:

Additional Editor Comments:

Please respond to all comments and concerns raised by the reviewers before submitting a revised version of the manuscript.

Reviewers' comments:

Reviewer's Responses to Questions

**Comments to the Author**

1. Is the manuscript technically sound, and do the data support the conclusions?

Reviewer #1: Yes

Reviewer #2: Yes

2. Has the statistical analysis been performed appropriately and rigorously? 

Reviewer #1: I Don't Know

Reviewer #2: Yes

3. Have the authors made all data underlying the findings in their manuscript fully available?

Reviewer #1: Yes

Reviewer #2: Yes

4. Is the manuscript presented in an intelligible fashion and written in standard English?

Reviewer #1: Yes

Reviewer #2: Yes

5. Review Comments to the Author

Reviewer #1: This retrospective study aims to develop high-performing Machine Learning and Deep Learning models in predicting hospital LoS while enhancing interpretability. The authors compare performance and interpretability of models trained only on structured tabular data with the same models trained only on unstructured clinical text data, and on mixed data.

Strong study overall with promising results, a few things to update/change

1. I think the introduction could be longer and provide more of a history of LoS prediction generally and why the field is moving toward machine learning and XAI ( with relevant refs) there is a recent review paper in PLOS Digital Health that will help you with this.

2. Why were these classifiers chosen for the first layer what was the motivation?

3. Clarify the following "Some “noise” remains in the output as reflected by meaningless tokens such as “1”. These are partly due to some inevitable “holes” in the preprocessing"

4. "However, there does not seem to be any consensus on the choice of the threshold" A binary classification might be of interest, but was a fine-grained classification tried? Why was this cut off decided? was there clnical input?

5. Any statistical significance testing performed?

6. Need to stress the fact that features that are collected are not routinely collected features so the study may struggle to generalise well

7. Discuss potential applications and implications of the models for healthcare professionals, administrators, and patients.

Reviewer #2: This is an article based on data from the publicly available MIMIC-III dataset, showing how a combination of methods based on structured and unstructured data can achieve excellent performance for the prediction of Length of Stay (LOS).

The rationale is well explained in the introduction.

The Machine Learning (ML) methods used in the article are cutting-edge, quite novel for the current application, and they seem to be appropriate. The quality of reporting is good.

The major limit is well ackownledged by the authors : the variables used to predict LOS span the entire stay and therefore are not available on admission, making it inpossible to predict LOS for newly admitted patients (which is the most interesting application). A variation of the algorithm could be devised by including only variables recorded on admission or during the first days, and it is expected that due to less available data such an algorithm would have lower performance.

Overall the article is very interesting, so I would consider this article suitable for publication after a major revision.

Major comments

Methods :

« The detailed parameters are given here: https://github.com/awslabs/autogluon/tree/master/tabular/src/autogluon/tabular/models »: this link redirects to the AutoGluon package source code, which could change on a daily basis. Therefore, the parameters indicated may have changed since the article was written. Instead, the authors should either :

- make a snapshot of the hyperparameters in the abovementionned repository as an addition to their own study page, or - extract the hyperparameters that were likely used and indicate them in an Appendix, or - acknowledge that these hyperparameters may not be available anymore due to the adaptive and rapidly moving nature of the software used.

Results :

Permutation Importance for urea is 100. It seems therefore that the authors have normalized raw permutation importance scores to transform them to percentages of the best performing feature. This could be reversed in order to have more information, or explained (as there may be different methods to calculate permutation importance).

Minor comments

Abstract

« are then merged into a tabular dataset » : the past tense needs to be used when describing what was done, the authors should replace « are » with « were »

Introduction :

« total number of LOS of all inpatients » : this is simply the « sum of LOS for all inpatients »

« as is LOS in general » : this can be removed to avoid some kind of unnecessary repetition.

Methods : treatment of missing data is well explained

The reported formula for Tukey’s upper fence should be 3 + 1.5 × (3 − 1)

(so the first « – » in the text should be «+ »). Nonetheless, it seems that the formula was correctly applied in the dataset ; the proportion of outliers that were identified looks reasonnable.

« There are three types of features selected as predictors in the dataset » : same as before, past tense should be used for specific study aspects : « were » instead of « are »

« Detailed descriptions of these variables are provided in Appendix 1. » : As this call is made in the Methods section the reader expects the Appendix to contain written information about how the variables were recorded, their definition etc. This call could be moved to the Results section.

The encoder used in Transformers is mentionned, but why not mention the decoder part ? This may or may not be intentional

The technical characteristics of the machine used could be provided, which GPUs if any, etc.

Results : for future reference, the unit of fit time should be provided (minutes, hours ?)

Older age seems to be associated with shorter LOS in the unadjusted bivariate analysis (Appendix 1), which is counterintuitive. In some European countries, older patients tend to have higher LOS as often there is no other solution for them in terms of post-acute care. Maybe this could be pointed out in the Discussion as another example of the difficulties of generalising the current findings for variable importance, and that each hospital should run their own models based on local data.

Discussion :

Other methods exist for the assessment of variable importance, such as LOCO (Leave One Covariate Out) (Reference : Lei J, G'Sell M, Rinaldo A, Tibshirani RJ, Wasserman L. Distribution-Free Predictive Inference For Regression. ArXiv; 2018. Available from: https://arxiv.org/abs/1604.04173), which was also probably also left out for computational reasons.

6. PLOS authors have the option to publish the peer review history of their article (what does this mean?). If published, this will include your full peer review and any attached files.

Reviewer #1: No

Reviewer #2: **Yes: **Jan Chrusciel

---

## [Author Response · Author response to Decision Letter 0]

27 Jun 2023

Response to the Editor and to the Reviewers

We are thankful to the editors and the reviewers for the opportunity to improve our article and to further consideration for publication. Following are the changes made to the manuscript.

Reviewers and Editors’ observations are in Calibri blue, our answers are in Times New Roman black. Changes made in the manuscript are in Calibri black, highlighted.

EDITOR

1. Please add to the abstract that your experiments were performed using AutoGluon if there is no novelty in the machine learning techniques (e.g., development of a new transformer) in your methods.

The abstract was modified to include mention of AutoGluon AutoML as the main software.

Change in manuscript (p. 15)

The study used the free and publicly available Medical Information Mart for Intensive Care (MIMIC) III database, on the open AutoML Library AutoGluon.

2. The text in figures 1-3 is blurred. Please replace them with figures of appropriate resolution.

We have generated new figures.

Figures in the pdf generated by submitting to PLOS ONE are indeed blurred. When clicking on the link in the same pdf “Click here to access/download Figure […]”, the quality appears to be good. We believe it might be an issue with how the website renders the submission pdf rather than an issue on our side, let us know how to proceed.

3. For Figures 2 and 3, instead of showing the absolute values of the weights, please show the important words in the top 20 positive and negative values.

We have implemented the Editor’s request, as illustrated in Figure 1 hereafter. Displaying the top positive and top negative features is a common practice when using LIME for local explanation, i.e. for individual instances (1,2). However when extending it to many instances, to get information on global (variable) explanation, the practice is to average the absolute value of the weights on an appropriate subset of the full dataset ⸺as illustrated in the Submodular Pick approach (3)⸺ or on all the dataset as we have done in this paper.

Why use the absolute value of the weights rather than the raw values when moving from local to global explanation?

Let us use an analogy. When measuring the overall error of a linear model using Ordinary Least Square, the overall error of prediction is measured either by the mean absolute error or the mean squared error (the error being the difference between the actual and predicted values of the outcome). Why not just take the aggregated average raw errors? Because the negative errors and the positive errors will cancel each other. The same issue arises for the coefficients: for some instances the same token will have a positive coefficient, and for others it will have negative coefficients. The averaging will thus erase the information contained in the data. The proof for this is in the following (see response to the editor & to the review - Figure 1).

When we examine the token <history> for lemmatized tokens, it is a mitigating factor for PLOS (negative coefficient); whereas <histori> for the stemmed tokens is an aggravating factor for PLOS (positive coefficient). Of course, this sign difference in coefficients may be explained by the difference in covariate tokens (and intercepts) between the two preprocessing approaches, however, when examining the two tokens in context, the most frequent words preceding and following each of them are very comparable (see response to the editor & to the review - Table 1). Hence, their having two opposite associations with the outcome is unwarranted.

The authors feel that including the averaged raw weight values in the main text can be not only confusing but also misleading for the readers. Nevertheless, we are happy to include it in a supplement material if the editor so sees fit.

Change in manuscript (p. 20)

A reliable interpretation, however, should include examination of keywords in context. For instance, the token “1” may appear as noise. Its presence may be explained by our choice of a light preprocessing, where we avoided removing numbers to preserve potentially important information (e.g. medication quantities). When looking at the most common keywords in the context of such token, we do in fact find a variety of medication-related words, such as “sig”, “mg”, “tablet”, “capsul”, “daili”, “po”, implying that Bio Clinical BERT utilizes it to spot medication frequency and/or dosage. It is important to consider that the LIME representations used here are based on linear (Lasso) approximations from two different models using each a different type of preprocessing (respectively lemmatization and stemming). Each token should therefore be interpreted in light of their covariate tokens. As shown in the Figure 3 (see response to the editor & to the review), the most important tokens in each model are not the same. This is compelling evidence that preprocessing matters and that each token should be interpreted within its context.

4. Please provide visual representations of attention (similar to figures 3 and 4 in "Attention Is All You Need", arXiv:1706.03762) to predict dwell time.

The authors made many Attention visualization attempts similar to the one suggested by the editor. Figure 4 provides an example of such an attempt.

Despite all our effort, we have not been able to generate a satisfactory visualization. Our current model uses a 512-sequence length, implying that we are considering long spans of texts. The attention of the [CLS] tokens used for the pooled output is thus spread out over all the tokens, and this does not support a clear and simple visual representation of attention over layers and heads (see response to the editor & to the review - Figure 4).

Regarding the paper referenced by the editor, the version published in NIPS (4) does not contain the figures in the arXiv:1706.03762 preprint. Moreover, the GitHub provided by the authors does not seem to have anything related to those visualization (https://github.com/jadore801120/attention-is-all-you-need-pytorch/tree/master), which is listed as a “ToDo”. Finally, in both the arXiv and NIPS versions, there is no mention nor explanation of those figures (besides their caption), and it is thus not clear from which dataset and model they are taking the example sentence from. More importantly, it seems that they are providing a toy example with a reduced sequence length, as the displayed sentences average roughly 25/30 words – whereas the discussed models within the paper go from 512 to 1024 sequence length. In sum, we have not found the illustration in the arXiv paper to be reproducible and our best attempt is provided in Figure 4 which is not really informative.

It is the authors' view that the option we have retained remains pertinent for the following reasons.

For a case like ours, we think that LIME explanations provide a more convenient and more readable view of what our model is focusing on to predict LOS, both locally at each instance level, and globally (when averaged over all instances), at a variable level. 

Moreover, the LIME weights aggregated over all documents highlight what tokens our model uses overall, contrary to a visualization of attention which would necessitate choosing a document, a layer and a head.

5. Please make the source code and relevant tutorials/examples available as part of an open-source repository on GitHub or GitLab. In addition, include a link to the repository in the manuscript.

The relevant change was made and the following text was added.

Change in manuscript (p. 26)

The rationale for binarizing the LOS output variable is explained and the code containing the whole preprocessing of the dataset along with all the code used in the study are openly available in the GitHub of the study (link: https://github.com/jaotombo/LOS_mixed_2022).

6. Clarify authorship (https://journals.plos.org/plosone/s/authorship)

The following text was added to clarify authorship.

Change in manuscript (p. 3)

Authors’ contributions:

 Franck Jaotombo: Conceptualization, Formal Analysis, Investigation, Methodology, Supervision, Writing (original draft, review & editing)

 Luca Adorni: Data curation, Formal Analysis, Software, Writing (original draft, review & editing)

 Badih Ghattas: Formal analysis, Writing (original draft, review & editing)

 Laurent Boyer: Resources, Supervision, Writing (original draft, review & editing)

 

REVIEWER # 1

1. I think the introduction could be longer and provide more of a history of LoS prediction generally and why the field is moving toward machine learning and XAI (with relevant refs) there is a recent review paper in PLOS Digital Health that will help you with this.

The Introduction has been extended, reorganized, and completed, including the reviewer's recommended reference, and others

Change in Manuscript (pp. 4-7)

Hospital length of stay (LOS) is defined as the time interval between hospital admission and discharge during a given admission event (5). As LOS enables a monitoring of the patients’ flows within the hospital’s care units and environment, it is considered as an indicator of resource consumption, cost and illness severity (5,6). Average length of stay (ALOS) is a macro indicator representing the average number of days patients spent in hospitals. It is the ratio between the sum of LOS for all inpatients in a year and the number of hospital stays, excluding day cases (7).

The ALOS in hospitals is also an indicator of efficiency in healthcare. Controlling for other factors, a shorter stay is likely to reduce the cost per stay and paves the way toward less expensive care settings. Longer stays suggest poor care coordination and may induce unnecessary in-hospital delays prior to rehabilitation or long-term care. Yet, some patients may be discharged too early when a longer hospital stay might have improved their conditions or reduced the likelihood of readmission. In 2019, the ALOS across the OECD countries was equal to 7.6 days (8).

One way to manage LOS is discharge planning. It is a customized individual plan designed for a patient, preparing the whole process leading to his leave after discharge, including the ongoing support in the community, and preventing readmission. Not only is discharge planning likely to reduce risks of readmission and improve patient satisfaction, it is especially instrumental in reducing LOS, thus significantly improving quality of care (9). Indeed, whereas discharge planning may include several aspects such as inputs from allied health staff, and discussions with community healthcare providers, some of its critical contributions rely on estimating discharge date and destination (DDD). Accurate prediction of DDD is directly based on the reliability of LOS prediction. Furthermore, not only do incorrect predictions jeopardize medical services and cause the dissatisfaction of patients and healthcare professionals, but they may also block and waste inpatient bed days. Conversely, accurate LOS prediction allows better resource allocation and care organization from patient admission to discharge preparation (10). Reliably predicting LOS could be an effective way to reduce costs and prevent unnecessary extended stays conducive to acquired infections, falls, overcrowding, or medical errors (11).

A recent systematic review proposed to categorize the approaches to predict LOS within three main groups. The first included methods based on statistical modeling such as the generalized linear models (linear and logistic regression); the second covered methods based on operational research, such as compartmental modeling, simulations, Markov models and phase-type distributions; the third were data mining and machine-learning-based methods (5). With the advent of the “big data” era and the rising interest on electronic health records (EHR), the machine learning approach is gaining more momentum. Bacchi and colleagues (12) argue that the assumption-free data-driven nature of machine learning would make it an optimal choice for reaching accurate prediction of LOS. 

Lequertier et al. (10) offer another extensive review on the methods used to predict LOS. While they highlight that LOS is still relevant in planning bed capacity, and discharge planning is still a current matter of concern in healthcare delivery, they also stress the difficulty in identifying an optimal method due to the diversity of data sources, input variables and metrics. These shortcomings of the current LOS research are, furthermore, highlighted by Stone et al. (5) : “(...) the performance of a given approach will vary depending on a large number of competing factors such as the number of patients a hospital admits, a patient’s diagnosis, the hospital’s urban/rural location, particular procedures or processes in place and care units, etc.” (p. 27), thus they suggest to work on models trained only on data systematically collected in the majority of hospitals. The authors stress equally the need to study the contribution of nursing admission data, given that the nurses spend much more time with the patients than the doctors, and are able to collect more information on the patients’ social background, home situation, lifestyle habits and overall livelihood constraints. Lequertier et al. (10) further recommend 1) a transparent restitution of population selection, data sources and input variables, handling of missing data, LOS transformations, and performance metrics; 2) avoiding arbitrarily excluding outliers which impairs validity; 3) using different datasets for training the model and testing the performance, and even avoiding the pitfall of splitting the data into overly optimistic or pessimistic datasets by using k cross-validation; 4) selecting metrics that account for the outcome distributions – especially in case of imbalanced datasets; 5) reporting the training time of the models; 6) using open and freely available datasets.

In clinical research, improving predictive performance is good but not nearly enough to encourage a wide adoption of ML models. Admittedly, the more sophisticated ML models such as Deep Learning (DL) may seem like black boxes (13,14), which clinicians and practitioners may find disconcerting as they expect more interpretability. Clinicians will most likely be reluctant to welcome the achievements of these models despite the benefits their predictive abilities might bring, as the derivation leading to their results comes with a poor explicit explanation, if any. Consequently, developing systems that support explainable and transparent decisions have become prevalent (15) as eXplainable Artificial Intelligence – XAI (16). Performance concerns the ability of a model to make correct predictions, while interpretability concerns to what degree the model allows for human understanding (17). Models exhibiting high performance are often more complex and less transparent, while interpretable models may be more limited in performance. Exploring the trade-off between performance and interpretability is one of the main goals of XAI (18,19). 

As LOS is a quantitative variable, several of the studies attempt to predict its value with Machine Learning (ML) regression models. Yet from the perspective of identifying patients at risk, predicting prolonged LOS (PLOS) may be the main concern as opposed to regular LOS (RLOS) (20). In such a case, the outcome to be predicted is categorical (binary) and the ML models to be used are classification models. This binarization process requires the choice of a cutoff point. However, there does not seem to be any consensus on the choice of the threshold (21): some select ad hoc cutoffs such as 7 days to obtain more balance datasets (22), others use statistical criteria such as the 75th, the 90th or 95th percentiles (20,23,24). It is therefore difficult to make a rigorous benchmark between the different studies predicting LOS (12). 

One way of improving the performance of LOS prediction is to resort to other data types such as medical imaging or free texts (clinical notes) (12). Free text may be collected from doctors’ and nurses’ clinical notes available in electronic health records (EHR), and leveraged to improve interpretability (1). Not only can clinical notes predict different types of outputs (25–27) but they may also increase the performance of the typical structured datasets in predicting LOS (22,28). Overall, their use may be a means of enhancing the trade-off between performance and interpretability (29)

In this article, we are exploring different ways of finding the best trade-off between performance and interpretability in LOS prediction by comparing results from models trained only on structured tabular data, with models trained only on unstructured clinical text data, and with models trained on mixed tabular structured and unstructured data - through data fusion.

2. Why were these classifiers chosen for the first layer what was the motivation?

For the sake of replicability, resource and time management, we relied as much as we could on the default settings of the AutoML AutoGluon platform. Most of the models used on the platform have been pre-tuned and the results obtained do not require further complex, advanced, time- and resource-consuming, hyperparameters tuning.

Change in Manuscript (p. 15)

Both static and dynamic structured data were merged as one structured tabular data and used to compare 14 ML models using AutoGluon TabularPredictor (30). The selected ML models cover a wide range of the most current, the most relevant, and best performing pre-tuned models available per default in AutoGluon:

3. Clarify the following "Some “noise” remains in the output as reflected by meaningless tokens such as “1”. These are partly due to some inevitable “holes” in the preprocessing"

Because LIME uses perturbed samples to build explainability from a penalized linear model (Lasso), some instability may occur (31), which may appear as irrelevant tokens amongst the most important variables. The token <1> may appear as one of such instances. However, probing more in-depth into its context, <1> is clearly not a noisy token but an indicator of quantity related to medication frequency or dosage. The manuscript was modified accordingly.

Change in Manuscript (p. 20)

A reliable interpretation, however, should include examination of keywords in context. For instance, the token “1” may appear as noise. Its presence may be explained by our choice of a light preprocessing, where we avoided removing numbers to preserve potentially important information (e.g., medication quantities). When looking at the most common keywords in the context of such token, we do in fact find a variety of medication-related words, such as “sig”, “mg”, “tablet”, “capsul”, “daili”, “po”, implying that Bio Clinical BERT utilizes it to spot medication frequency and/or dosage. It is important to consider that the LIME representations used here are based on linear (Lasso) approximations from two different models using each a different type of preprocessing (respectively lemmatization and stemming). Each token should therefore be interpreted in light of their covariate tokens. As shown in the Figure 3, the most important tokens in each model are not the same. This is compelling evidence that preprocessing matters and that each token should be interpreted within its context.

4. "However, there does not seem to be any consensus on the choice of the threshold." A binary classification might be of interest, but was a fine-grained classification tried? Why was this cut-off decided? Was there clinical input?

This cut-off was decided based on several reasons:

1 - statistical: we looked at the distribution of LOS which displays a huge peak and a flat part. Therefore, there is no statistical reason to select more than two categories (peak vs. flat). 

2 - historical, most of the studies on LOS use either regression or binary classification. 

3 - clinical, what is costly are the prolonged LOS (PLOS) so we want to prevent those by predicting their occurrence. We assume that in OECD countries, prolonged LOS are rare. It should certainly not be 50% of the stays. In statistical terms, rare implies outliers, and the simplest way of computing outliers without making any distribution assumption is the Tukey's fences formula used here.

Change in Manuscript (p. 8)

This cut-off was first chosen for a statistical reason. The distribution of the LOS is made of one narrow peak followed by a flat line of outliers, suggesting a binary distribution (Appendix 4). It is also justified for a historical reason: most of the studies on LOS use either regression or binary classification. Lastly, it is founded on public health reasoning. We assume that in OECD countries, PLOS are rare, certainly much less than 50% of the stays. In statistical terms, rare may translate as outliers, and the simplest way of computing outliers without making any distribution assumption is the Tukey's fences formula used here. Our study amounts therefore to a binary classification problem where the positive class represents the prolonged stays (PLOS = 7.28%) vs. the regular stays (RLOS = 92.72%).

5. Any statistical significance testing performed?

1- Statistical tests could have been implemented to investigate the significance of the difference in performance. A reliable method would require resampling the training set 100 times, then train each ML model and compute the value of each metric on the validation set for each resampling. This would provide a vector of 100 values for each metric and for each ML model (4 metrics x 14 models x 100 resamplings). These vectors may then be used for mean comparison (T tests) between each ML models – for more details see (32).

However, even if we were to do only 30 resamplings, given the complexity of the models, it would be considerably time- and resource-consuming.

2- The k-fold bagging applied by AutoGluon, already outputs an averaged performance that smooths up the values and eliminates much dispersion. Given the sample size used to train the data (more than 24k rows), we would not expect any significant variation in performance from one resampling to another.

Given the little potential added value compared to the required energy investment, we opted to avoid further statistical testing.

6. Need to stress the fact that features that are collected are not routinely collected features so the study may struggle to generalise well

1- If the goal is generalizability, then one should favor those variables that are primarily systematically or routinely collected in most hospitals. Ours is very specific to the Boston BIDMC and to the intensive care units, thus it will not generalize well (5).

2- We may argue, however, that the goal may not be necessary to generalize the variables from hospital to hospital, but to generalize the process. Indeed, keeping only those variables routinely collected in most hospitals will reduce performance and interpretability as it will not include relevant variables that are specific to each institution and conducive to greater performance and interpretability. In a data mining and machine learning approach, it may be more pertinent to include all relevant data in the model in order to maximize performance and interpretability. These variables may be different from one hospital to another but what is generalized is the fact that performance and interpretability are maximized.

Change in Manuscript (p. 30)

This study is not without limitations. The MIMIC III database is used here in a retrospective study. In real life and in real time, many of these variables will not always be available, thereby questioning the generalizability of our results. For the sake of generalizability one should favor those variables that are primarily, systematically or routinely collected in most hospitals (5,33). Our results are very specific to the Boston Beth Israel Deaconess Medical Center with a focus on intensive care units; thus it may not generalize well. Yet, keeping only those variables routinely collected in most hospitals will reduce performance and interpretability as it will not include relevant variables that are specific to each institution and conducive to greater performance and interpretability. The specificity of each hospital may be accounted for through usage of ready to use models retrained on each local site data through threshold adjustment and transfer learning (34). One may argue that if generalizability may be a priority for research, including all pertinent data in the model as to maximize performance and interpretability may be the priority for the practitioners and the managers. Indeed some authors recommend to embrace a wider view of generalizability where the goal is to focus on broader questions about when, how, and why ML systems have clinical utility and ensure that these systems work as intended for both clinicians and patients (35).

7. Discuss potential applications and implications of the models for healthcare professionals, administrators, and patients.

Discussion was expanded to better explain potential implications for healthcare professionals, administrators and patients.

Change in Manuscript (p. 29)

There may be several applications to models like ours. They may be utilized as tools to aid making a precise diagnosis leading to highly desirable personalization of patients’ management (36). Better adapted to big data than the conventional statistical models, they may scale to include up to billions of patients’ records, and use a single, distributed patient representation – from different data sources such as EHRs, genomics, social activities and other features describing individual status. Deployed into a healthcare system, these models would be constantly updated to follow the changes in patient population and will support clinicians in their daily activities (37). Another area where these models may have comprehensive leverage is in healthcare operations management. ML models based on weak learners such as in boosted models or in ensemble learning models have shown to be quite relevant in predicting workflow events as well as in identifying key operational features (33). Their efficacy is substantiated by acknowledging that any outcome of a clinical workflow is influenced by a plethora of different factors, and each of them can be considered as a weak learner due to their little impact on the outcome. As an illustration, one boosted ML models, deployed on an information system and trained in real time was used to predict waiting time in a facility and hailed by the patients (33). A different display was also made available and customized as an administrator view for the facility manager, allowing the staff to examine gaps between the actual and the predicted values, and providing the means to investigate new features to be used for improvement. As the performance of the models reach a satisfactory level, feature selection such as retaining the most important features were applied to determine the key factors contributing to the operational outcome - e.g., time delay in the creation of radiology reports (33). It is not too much of a stretch to envision how these different applications would be enhanced – in terms of performance and explainability – if fused structured and unstructured data were used to train the ML models. It would improve the patients’ journey, support the practitioners in their monitoring and caring tasks, and facilitate the (resources) planning and management of the facilities.

 

REVIEWER # 2

1- « The detailed parameters are given here: https://github.com/awslabs/autogluon/tree/master/tabular/src/autogluon/tabular/models »: this link redirects to the AutoGluon package source code, which could change on a daily basis. Therefore, the parameters indicated may have changed since the article was written. Instead, the authors should either :- make a snapshot of the hyperparameters in the above-mentioned repository as an addition to their own study page, or - extract the hyperparameters that were likely used and indicate them in an Appendix, or - acknowledge that these hyperparameters may not be available anymore due to the adaptive and rapidly moving nature of the software used.

We expanded and clarified parameter tuning in the Method section. We provided the version of AutoGluon used in the paper together with a link to where past versions of AutoGluon can be downloaded, so that anyone can correctly replicate our results or peek into the actual parameters.

Change in Manuscript (p. 16)

Hyperparameter tuning: the AutoGluon platform provides sophisticated means of tuning the hyperparameters. However, given the large number of models to be trained, the already satisfactory level of performance with the default parameter values, and the goal of our study, we have reduced this part to the bare minimum i.e., the choice of the evaluation metrics in tuning: the ROC AUC. We used version 0.4.1 of AutoGluon. Both parameters and models may have changed and improved over time due to the high frequency of new releases from the AutoGluon team. To properly replicate our results or check the hyperparameters in detail, the correct older version of the package must be downloaded from PyPi (https://pypi.org/project/autogluon/#history). In principle, the latest version of AutoGluon should nonetheless lead to very close if not identical results.

2- Permutation Importance for urea is 100. It seems therefore that the authors have normalized raw permutation importance scores to transform them to percentages of the best performing feature. This could be reversed in order to have more information, or explained (as there may be different methods to calculate permutation importance).

We have included both permutation importance figures based on raw values and on values normalized on the most important feature.

3- « are then merged into a tabular dataset » : the past tense needs to be used when describing what was done, the authors should replace « are » with « were »

The past tense was corrected in the paper and in the abstract.

4- « total number of LOS of all inpatients » : this is simply the « sum of LOS for all inpatients »

This was also modified.

5- « as is LOS in general » : this can be removed to avoid some kind of unnecessary repetition.

This was removed altogether from the paper.

6- The reported formula for Tukey’s upper fence should be 3 + 1.5 × (3 − 1). (so the first « – » in the text should be «+ »). Nonetheless, it seems that the formula was correctly applied in the dataset ; the proportion of outliers that were identified looks reasonable.

This was indeed a typo. The code is correct and has been double-checked.

7- « There are three types of features selected as predictors in the dataset » : same as before, past tense should be used for specific study aspects : « were » instead of « are »

Fixed the past tense.

8- « Detailed descriptions of these variables are provided in Appendix 1. » : As this call is made in the Methods section the reader expects the Appendix to contain written information about how the variables were recorded, their definition, etc. This call could be moved to the Results section.

Appendix 1 was removed and the tables were shifted to the main section under "Study Features", all the tables' and appendices' numbering were updated.

9- The encoder used in Transformers is mentioned, but why not mention the decoder part ? This may or may not be intentional.

For a classification task, Transformers only require the encoder part. The decoder is used for text generation. As our research is limited to a classification task, we have intentionally omitted to mention the decoder which is beyond the scope of the study.

10- The technical characteristics of the machine used could be provided, which GPUs if any, etc.

All the models were trained on Google Colab Pro+. We added a small paragraph in the paper explaining it and clarifying the characteristics of the machines used.

Change in Manuscript (p. 17)

All models have been trained using Google Colab Pro+ with GPU enabled machines. Google Colab assigns a type of machine every time a new notebook is initialized, but may switch to other types. Examples of machine used are: V100 (GPU RAM: 16GB; CPUs: 2 vCPU, up to 52 GB of RAM); P100 (GPU RAM: 16 GB; CPUs: 2 vCPU, up to 25 GB of RAM); T4 (GPU RAM: 16 GB; CPUs: 2 vCPU, up to 25 GB of RAM). 

11- Results : for future reference, the unit of fit time should be provided (minutes, hours ?)

Time is in seconds. Table 2 was updated accordingly.

12- Older age seems to be associated with shorter LOS in the unadjusted bivariate analysis (Appendix 1 now Table 1), which is counterintuitive. In some European countries, older patients tend to have higher LOS as often there is no other solution for them in terms of post-acute care. Maybe this could be pointed out in the Discussion as another example of the difficulties of generalising the current findings for variable importance, and that each hospital should run their own models based on local data.

Indeed, a recent study on LOS conducted in a University Hospital in the South of France (32) indicates longer LOS for older patients. Our data are essentially based on ICU patients from the Boston Beth Israel Deaconess Center, which is a very different type of institution. This would be the most obvious possible explanation.

By examining the bivariate statistics (chi-square tests) between several of the categorical variables, we note that:

1- There are indeed significantly fewer PLOS admissions for the 65-84 and 85+ years categories than would be otherwise expected under hypothesis of independence.

2- There are significantly more Emergency admissions for the 18-44 and 85 years categories, but significantly less Emergency admissions in the 45-64 and 65-84 years categories than expected.

3- There are significantly more Elective (planned) admissions for the 45-64 and 65-84 years categories, but significantly less Elective admissions in the 18-44 and 85+ years categories than expected

4- There are significantly more PLOS admissions in the Emergency and Urgent admissions, and significantly less Elective (Planned) admissions than expected.

To summarize: most PLOS are due to Emergency or Urgent admissions and concern the 18-44 years and 85+ years, which are smaller in proportion.

The 45-64 years and 65-84 years categories are mostly in the planned admissions, thus more likely to be in the RLOS category, and larger in proportion.

13- Other methods exist for the assessment of variable importance, such as LOCO (Leave One Covariate Out) (Reference : Lei J, G'Sell M, Rinaldo A, Tibshirani RJ, Wasserman L. Distribution-Free Predictive Inference For Regression. ArXiv; 2018. Available from: https://arxiv.org/abs/1604.04173), which was also probably also left out for computational reasons.

There are many books and articles written on eXplainable AI, of which variable importance is but one technique amongst many others (38,39). We cannot cover them all, and have made our choice mostly for computational reasons and for convenience. Indeed, most of the alternative algorithms to ours require much longer computation time.

LOCO (Lei et al.,2017) is most likely a better alternative as it removes the noise generated by the permutated variable but not very convenient since available only on R. SHAP (Lundberg & Lee, 2017) is also another very interesting alternative but more computationally intensive than LIME. Thus, we have settled for permutation importance and LIME.

Change in Manuscript (p. 14)

There are many resources available on XAI (38), including methods to estimate variable importance such as the Leave One Covariate Out (40). In this study we have focused mostly on the overall importance of the 20 most relevant features, either through permutation importance (41) or through Local Interpretable Model-agnostic Explanations (LIME) (42). In the latter case, for each feature, we computed the value of its local contribution on predicting each instance, then averaged these over the whole dataset. 

References

1. Mardaoui D, Garreau D. An Analysis of LIME for Text Data. In: Proceedings of The 24th International Conference on Artificial Intelligence and Statistics [Internet]. PMLR; 2021 [cité 26 mai 2023]. p. 3493‑501. Disponible sur: https://proceedings.mlr.press/v130/mardaoui21a.html

2. Visani G, Bagli E, Chesani F, Poluzzi A, Capuzzo D. Statistical stability indices for LIME: obtaining reliable explanations for Machine Learning models. J Oper Res Soc. 2 janv 2022;73(1):91‑101. 

3. Ribeiro MT, Singh S, Guestrin C. « Why Should I Trust You? »: Explaining the Predictions of Any Classifier. In: Proceedings of the 22nd ACM SIGKDD International Conference on Knowledge Discovery and Data Mining [Internet]. New York, NY, USA: Association for Computing Machinery; 2016 [cité 24 juin 2023]. p. 1135‑44. (KDD ’16). Disponible sur: https://dl.acm.org/doi/10.1145/2939672.2939778

4. Vaswani A, Shazeer N, Parmar N, Uszkoreit J, Jones L, Gomez AN, et al. Attention is All you Need. In: Advances in Neural Information Processing Systems [Internet]. Curran Associates, Inc.; 2017 [cité 18 juin 2022]. Disponible sur: https://proceedings.neurips.cc/paper/2017/hash/3f5ee243547dee91fbd053c1c4a845aa-Abstract.html

5. Stone K, Zwiggelaar R, Jones P, Parthaláin NM. A systematic review of the prediction of hospital length of stay: Toward a unified framework. PLOS Digit Health. 14 avr 2022;1(4):e0000017. 

6. Chang KC, Tseng MC, Weng HH, Lin YH, Liou CW, Tan TY. Prediction of Length of Stay of First-Ever Ischemic Stroke. Stroke. nov 2002;33(11):2670‑4. 

7. OECD. Health at a Glance 2019: OECD Indicators [Internet]. Paris: Organisation for Economic Co-operation and Development; 2019 [cité 21 oct 2022]. Disponible sur: https://www.oecd-ilibrary.org/fr/social-issues-migration-health/health-at-a-glance-2019_4dd50c09-en

8. OECD. Health at a Glance 2021: OECD Indicators [Internet]. Paris: Organisation for Economic Co-operation and Development; 2021 [cité 21 oct 2022]. Disponible sur: https://www.oecd-ilibrary.org/fr/social-issues-migration-health/health-at-a-glance-2021_ae3016b9-en

9. Bacchi S, Gluck S, Tan Y, Chim I, Cheng J, Gilbert T, et al. Prediction of general medical admission length of stay with natural language processing and deep learning: a pilot study. Intern Emerg Med. sept 2020;15(6):989‑95. 

10. Lequertier V, Wang T, Fondrevelle J, Augusto V, Duclos A. Hospital Length of Stay Prediction Methods: A Systematic Review. Med Care. 1 oct 2021;59(10):929‑38. 

11. Simmons FM. CEU: Hospital overcrowding: An opportunity for case managers. Case Manag. 1 juill 2005;16(4):52‑4. 

12. Bacchi S, Tan Y, Oakden-Rayner L, Jannes J, Kleinig T, Koblar S. Machine Learning in the Prediction of Medical Inpatient Length of Stay. Intern Med J [Internet]. 2020 [cité 4 juill 2021]; Disponible sur: http://onlinelibrary.wiley.com/doi/abs/10.1111/imj.14962

13. Guidotti R, Monreale A, Ruggieri S, Turini F, Giannotti F, Pedreschi D. A Survey of Methods for Explaining Black Box Models. ACM Comput Surv. août 2018;51(5):93:1-93:42. 

14. Mahmoudi E, Kamdar N, Kim N, Gonzales G, Singh K, Waljee AK. Use of electronic medical records in development and validation of risk prediction models of hospital readmission: systematic review. BMJ. 8 avr 2020;369:m958. 

15. Holzinger A, Biemann C, Pattichis CS, Kell DB. What do we need to build explainable AI systems for the medical domain? [Internet]. arXiv; 2017 [cité 8 juin 2023]. Disponible sur: http://arxiv.org/abs/1712.09923

16. Barredo Arrieta A, Díaz-Rodríguez N, Del Ser J, Bennetot A, Tabik S, Barbado A, et al. Explainable Artificial Intelligence (XAI): Concepts, taxonomies, opportunities and challenges toward responsible AI. Inf Fusion. 1 juin 2020;58:82‑115. 

17. Johansson U, Sönströd C, Norinder U, Boström H. Trade-off between accuracy and interpretability for predictive in silico modeling. Future Med Chem. avr 2011;3(6):647‑63. 

18. Linardatos P, Papastefanopoulos V, Kotsiantis S. Explainable AI: A Review of Machine Learning Interpretability Methods. Entropy. janv 2021;23(1):18. 

19. Lundberg SM, Erion G, Chen H, DeGrave A, Prutkin JM, Nair B, et al. From local explanations to global understanding with explainable AI for trees. Nat Mach Intell. janv 2020;2(1):56‑67. 

20. Marfil-Garza BA, Belaunzarán-Zamudio PF, Gulias-Herrero A, Zuñiga AC, Caro-Vega Y, Kershenobich-Stalnikowitz D, et al. Risk factors associated with prolonged hospital length-of-stay: 18-year retrospective study of hospitalizations in a tertiary healthcare center in Mexico. PLOS ONE. 8 nov 2018;13(11):e0207203. 

21. Williams TA, Ho KM, Dobb GJ, Finn JC, Knuiman M, Webb SAR. Effect of length of stay in intensive care unit on hospital and long-term mortality of critically ill adult patients. Br J Anaesth. 1 avr 2010;104(4):459‑64. 

22. Chrusciel J, Girardon F, Roquette L, Laplanche D, Duclos A, Sanchez S. The prediction of hospital length of stay using unstructured data. BMC Med Inform Decis Mak [Internet]. 2021 [cité 21 juin 2022];21. Disponible sur: https://journals.scholarsportal.info/details/14726947/v21inone/nfp_tpohlosuud.xml

23. Blumenfeld YJ, El-Sayed YY, Lyell DJ, Nelson LM, Butwick AJ. Risk Factors for Prolonged Postpartum Length of Stay Following Cesarean Delivery. Am J Perinatol. juill 2015;32(9):825‑32. 

24. Collins TC, Daley J, Henderson WH, Khuri SF. Risk Factors for Prolonged Length of Stay After Major Elective Surgery. Ann Surg. août 1999;230(2):251. 

25. Huang K, Altosaar J, Ranganath R. ClinicalBERT: Modeling Clinical Notes and Predicting Hospital Readmission. ArXiv190405342 Cs [Internet]. 28 nov 2020 [cité 3 mars 2022]; Disponible sur: http://arxiv.org/abs/1904.05342

26. Orangi-Fard N, Akhbardeh A, Sagreiya H. Predictive Model for ICU Readmission Based on Discharge Summaries Using Machine Learning and Natural Language Processing. Informatics. mars 2022;9(1):10. 

27. Teo K, Yong CW, Chuah JH, Murphy BP, Lai KW. Discovering the Predictive Value of Clinical Notes: Machine Learning Analysis with Text Representation. J Med Imaging Health Inform. 1 déc 2020;10(12):2869‑75. 

28. Zhang D, Yin C, Zeng J, Yuan X, Zhang P. Combining structured and unstructured data for predictive models: a deep learning approach. BMC Med Inform Decis Mak. 29 oct 2020;20(1):280. 

29. Shickel B, Tighe PJ, Bihorac A, Rashidi P. Deep EHR: A Survey of Recent Advances in Deep Learning Techniques for Electronic Health Record (EHR) Analysis. IEEE J Biomed Health Inform. sept 2018;22(5):1589‑604. 

30. Erickson N, Mueller J, Shirkov A, Zhang H, Larroy P, Li M, et al. AutoGluon-Tabular: Robust and Accurate AutoML for Structured Data [Internet]. arXiv; 2020 [cité 3 juill 2022]. Disponible sur: http://arxiv.org/abs/2003.06505

31. Alvarez-Melis D, Jaakkola TS. On the Robustness of Interpretability Methods [Internet]. arXiv; 2018 [cité 20 juin 2022]. Disponible sur: http://arxiv.org/abs/1806.08049

32. Jaotombo F, Pauly V, Fond G, Orleans V, Auquier P, Ghattas B, et al. Machine-learning prediction for hospital length of stay using a French medico-administrative database. J Mark Access Health Policy. 31 déc 2023;11(1):2149318. 

33. Pianykh OS, Guitron S, Parke D, Zhang C, Pandharipande P, Brink J, et al. Improving healthcare operations management with machine learning. Nat Mach Intell. mai 2020;2(5):266‑73. 

34. Yang J, Soltan AAS, Clifton DA. Machine learning generalizability across healthcare settings: insights from multi-site COVID-19 screening. Npj Digit Med. 7 juin 2022;5(1):1‑8. 

35. Futoma J, Simons M, Panch T, Doshi-Velez F, Celi LA. The myth of generalisability in clinical research and machine learning in health care. Lancet Digit Health. 1 sept 2020;2(9):e489‑92. 

36. Ashton JJ, Young A, Johnson MJ, Beattie RM. Using machine learning to impact on long-term clinical care: principles, challenges, and practicalities. Pediatr Res. janv 2023;93(2):324‑33. 

37. Miotto R, Wang F, Wang S, Jiang X, Dudley JT. Deep learning for healthcare: review, opportunities and challenges. Brief Bioinform. 27 nov 2018;19(6):1236‑46. 

38. Mehta M, Palade V, Chatterjee I. Explainable Ai: Foundations, Methodologies and Applications. 1st ed. 2023 édition. Springer International Publishing AG; 2022. 256 p. 

39. Molnar C. Interpretable Machine Learning [Internet]. 2022 [cité 14 juin 2022]. Disponible sur: https://christophm.github.io/interpretable-ml-book/

40. Lei J, G’Sell M, Rinaldo A, Tibshirani RJ, Wasserman L. Distribution-Free Predictive Inference For Regression [Internet]. arXiv; 2017 [cité 13 juin 2023]. Disponible sur: http://arxiv.org/abs/1604.04173

41. Altmann A, Toloşi L, Sander O, Lengauer T. Permutation importance: a corrected feature importance measure. Bioinformatics. 15 mai 2010;26(10):1340‑7. 

42. Garreau D, Luxburg U. Explaining the Explainer: A First Theoretical Analysis of LIME. In: Proceedings of the Twenty Third International Conference on Artificial Intelligence and Statistics [Internet]. PMLR; 2020 [cité 20 juin 2022]. p. 1287‑96. Disponible sur: https://proceedings.mlr.press/v108/garreau20a.html

---

## [Decision Letter · Decision Letter 1]

26 Jul 2023

Finding the best trade-off between performance and interpretability in predicting Hospital Length of Stay using Structured and Unstructured Data

PONE-D-23-09830R1

Dear Dr. Jaotombo,

We’re pleased to inform you that your manuscript has been judged scientifically suitable for publication and will be formally accepted for publication once it meets all outstanding technical requirements.

Kind regards,

Takuma Shibahara

Academic Editor

PLOS ONE

Additional Editor Comments (optional):

Reviewers' comments:

Reviewer's Responses to Questions

**Comments to the Author**

1. If the authors have adequately addressed your comments raised in a previous round of review and you feel that this manuscript is now acceptable for publication, you may indicate that here to bypass the “Comments to the Author” section, enter your conflict of interest statement in the “Confidential to Editor” section, and submit your "Accept" recommendation.

Reviewer #1: All comments have been addressed

Reviewer #2: All comments have been addressed

2. Is the manuscript technically sound, and do the data support the conclusions?

Reviewer #1: Yes

Reviewer #2: Yes

3. Has the statistical analysis been performed appropriately and rigorously? 

Reviewer #1: Yes

Reviewer #2: Yes

4. Have the authors made all data underlying the findings in their manuscript fully available?

Reviewer #1: Yes

Reviewer #2: Yes

5. Is the manuscript presented in an intelligible fashion and written in standard English?

Reviewer #1: Yes

Reviewer #2: Yes

6. Review Comments to the Author

Reviewer #1: I want to commend the authors for addressing all of my previous comments with great attention to detail.

The changes you've made have not only enhanced the clarity and coherence of the paper but also significantly strengthened its contributions to the field. You've demonstrated an excellent understanding of the topic, and it's clear that considerable time and effort went into this manuscript. Your study has potential to influence future work and inspire continued exploration in this area.

I look forward to seeing it published and anticipate that it will be well-received by the broader academic community.

Reviewer #2: (No Response)

7. PLOS authors have the option to publish the peer review history of their article (what does this mean?). If published, this will include your full peer review and any attached files.

Reviewer #1: No

Reviewer #2: **Yes: **Jan Chrusciel

---

## [Editor Report · Acceptance letter]

10 Aug 2023

PONE-D-23-09830R1 

Finding the best trade-off between performance and interpretability in predicting Hospital Length of Stay using Structured and Unstructured Data 

Dear Dr. Jaotombo:

I'm pleased to inform you that your manuscript has been deemed suitable for publication in PLOS ONE. Congratulations! Your manuscript is now with our production department. 

Kind regards, 

on behalf of

Dr. Takuma Shibahara 

Academic Editor

PLOS ONE